# Two-year-olds' visual exploration of response options during memory decisions predicts metamemory monitoring one year later

Sarah Leckey [1,2] ✉, Diana Selmeczy [3] & Simona Ghetti [1,2] ✉

Introspection on memory states guides decision-making, but little is known about how it emerges in childhood. Toddlers' behavioral responses to difficult memory decisions (e.g., information seeking) suggest early capacity to track uncertain situations, but it is unclear whether these behaviors relate to later emerging capacity to introspect on memory accuracy (i.e., metamemory monitoring). In a pre-registered longitudinal study, 176 25- to 34-month-olds encode images, then are asked to select the familiar image from arrays that also include a new image (Time 1). One year later (Time 2), 157 participants complete a similar memory task and report decision confidence. Higher gaze transitions between responses, indicative of evaluation processes, faster response latencies, and greater memory at Time 1 predict Time 2 metamemory monitoring (i.e., greater confidence for accurate than inaccurate decisions). At Time 2, gaze transitions are associated with lower overall confidence. Overall, this research reveals potential building blocks of emerging metamemory monitoring.

The ability to reflect on memory accuracy, or metamemory monitoring[1], plays a critical role in regulating learning and decision-making[2]. For example, if students feel uncertain about the material covered in an upcoming test, they may decide to ask their teacher for help or spend more time studying[3]. Recognizing a lack of confidence in one's memory may also prevent eyewitnesses from making wrongful incriminating allegations[4]. Several studies have now demonstrated that even preschoolers show evidence of metamemory monitoring (i.e., indicated by greater subjective confidence for accurate compared to inaccurate memory decisions[5]), but metamemory monitoring continues to improve during the elementary school years[6] and beyond[7]. Its developmental origin, however, remains unclear, but this understanding would be critical to gain insight on how young children begin to reflect on and regulate their learning and memory decisions.

A handful of studies have reported that infants and toddlers' behaviors suggest an early origin of metacognition. For example, 20-month-olds are more likely to turn to their caregivers for help during difficult compared to easy memory decisions[8], and 2.5-year-olds are more likely to ask for help after they made an inaccurate compared to an accurate memory decision[9]. Although these behaviors suggest a nascent ability to evaluate memory states, it is not clear whether and how they capture processes that pave the way for children's later emerging capacity to evaluate and report on their memory states. An answer to this question would inform ongoing debates concerning the functional significance of these early behaviors. One hypothesis is that these early behaviors reflect a rudimentary metacognitive system that is already in place in infancy and allows for likely automatic, unconscious, monitoring and control of actions. This rudimentary skill would become more sophisticated over time and develop into the conscious version of metacognition once children achieve the necessary abilities such as language[10,11]. Several studies in infants, utilizing word learning paradigms, have shown

[1]Center for Mind and Brain, University of California, Davis, Davis, CA, USA. [2]Department of Psychology, University of California, Davis, Davis, CA, USA. [3]Department of Psychology, University of Colorado, Colorado Springs, Colorado Springs, CO, USA. ✉e-mail: ssleckey@ucdavis.edu; sghetti@ucdavis.edu

evidence for this rudimentary metacognitive system[12,13]. For example, when 12-month-olds are asked to find a novel toy by name and are, by definition, ignorant about the correct choice, they pay closer attention to the adult who has reliably given correct answers in the past, indicating an early ability to recognize when information seeking is needed[12]. Although it is not clear whether these rudimentary skills are in themselves metacognitive (i.e., whether infants have conscious access to their knowledge or ignorance states), or are precursors of metacognitive skills, the prediction is that there would be a developmental relation between earlier skills and later metamemory skills. An alternative hypothesis is that these early behaviors reflect a more general tendency to appraise risk and choose the less risky option to avoid goal thwarting[14]. From this perspective, conscious awareness and conceptual understanding of mentalizing operations is necessary to classify a process as metacognitive and therefore these early behaviors would not necessarily reflect metamemory and/or predict later developing metamemory skills.

Open questions on the emergence of metamemory monitoring stem primarily from a focus on other processes concerning early memory functioning. Although a large body of research has revealed that even infants and toddlers have remarkable memory abilities including a capacity to form durable memories[15] that can be reinstated from multiple cues[16], research has not traditionally examined how young children gain awareness of their memory states in order to regulate their decisions. Indeed, most previous studies put considerable effort into minimizing response demands and decision processes to facilitate young children's demonstration of what they remember, unencumbered by effortful, potentially strategic, processes that could interfere with the production of the relevant behavior (i.e., looking preferences[17], imitation[18], conditioned responses[19]).

The approach to minimize these processes during memory decisions seems to have been well-founded. For example, a recent study demonstrated just how difficult memory decisions are for young children, but also how informative young children's eye movements may be during decision-making. Leckey and colleagues[20] had 2-year-olds complete a memory task under two retrieval conditions, after an encoding phase which involved viewing a series of individual object images (Experiment 1 and its replication). During retrieval, the toddlers were asked to either simply look at stimuli consisting of a new and familiar image (Passive condition) or to select the familiar image by pointing (Active condition). Looking preferences towards the novel items were collected in both the Passive and Active conditions. Overall, toddlers' looking preferences during the Passive condition showed they clearly remembered the studied images (i.e., showed high novelty preferences). However, accuracy levels were at chance during the Active condition, as measured by their response selection, underscoring toddlers' difficulty in using memory signals to guide overt response selections. Although accuracy was low at the group level, a subset of toddlers achieved high levels of accuracy in the Active condition. This high-performing group showed similar levels of novelty preference as the low-performing group in the Passive condition when no response was required, suggesting that the groups were comparable in memory retention. However, the high-performing group distributed their attention more equally between the response options during the Active condition. This suggested that toddlers were more likely to successfully select the correct responses if they overcame the tendency to examine the novel item and engaged in a visual comparison of the response options. This memory assessment at 2 years of age provided the initial wave of data for the current longitudinal study.

In another study, two-year-olds shifted their gaze more frequently in more difficult compared to easier perceptual decisions[21] and adults have been shown to report lower confidence for trials in which they shifted their gaze more often between response options[22]. Thus, the examination of eye movements provides a powerful means to gain insight not only on the content of memory representations through

assessing novelty preferences, but also on how gaze transitions prior to rendering a decision may inform subjective confidence. Therefore, we asked whether gaze transitions between response options represent a process of memory evidence collection which may provide the foundation for the capacity to subjectively assess one's memory accuracy.

Eye movements may not be the only behavioral indicator of an early emerging capacity of metamemory monitoring. Older children and adults have been shown to report lower confidence scores if they take longer to respond across cognitive domains in self-paced tests[23–25]. When it comes to memory, studies examining early manifestations of metamemory[8,9] have suggested that infants' bids for help are accompanied with hesitation, and possibly (although not assessed) slower responses, justifying a formal examination of this behavioral indicator. However, Koriat and Ackerman[23,26] reported that response latency is a more sensitive cue to confidence in older compared to younger children. Indeed, in their work, there were age-related differences in the reliance of children on response latencies for difficult decisions, with the relation between response latencies and metamemory monitoring being lower in 7-year-olds compared to 11-year-olds, casting doubt as to whether toddlers may already use it when they commit to memory decisions. Moreover, models that formally examine the relation between accuracy, response latencies, and confidence (i.e., two-stage dynamic signal detection theory; 2DSD[27]) have shown that with tasks in adults that emphasize speed over accuracy, inaccurate responses are made more quickly and given lower confidence responses, compared to accurate responses. These findings make it even more difficult to derive precise predictions in toddlers who may not prioritize memory accuracy over speed as older children and adults might do on self-paced tasks and might favor providing fast responses. Overall, previous research suggests that gaze transitions and response latencies may signal early assessments of memory accuracy, leading to the possibility that these early behavioral responses may be precursors of metamemory monitoring and in line with the idea that a rudimentary form of metamemory monitoring may be present in children as young as 2 years of age.

We conducted a pre-registered longitudinal study (Open Science Framework https://osf.io/9wz2m) to examine how individual differences in gaze transitions between response options and response latencies at the age of 2 predicted metamemory monitoring abilities, measured through subjective assessments of memory confidence at the age of 3. Metamemory ability was measured as the difference in confidence between accurate and inaccurate memory decisions. At the first assessment point (Time 1), 2-year-olds completed a memory task in which they were asked to identify a previously seen image from an array consisting of one old and one new image (Active Condition[20]; Fig. 1a). We additionally utilized results from a novelty preference task at Time 1 (Passive Condition[20]) to explore whether any relation found between gaze transitions at Time 1 and metamemory monitoring at Time 2 was contingent on the toddlers making an overt memory decision as opposed to demonstrating any evidence of memory retention.

These children returned approximately one year later to complete their second assessment (Time 2). At that time, they completed a similar memory task which also required them to identify the old item in an array including one old and one new image and additionally probed subjective confidence ratings for each memory decision (e.g., Fig. 1b[5]). At each time point, participants completed two parallel versions of the task, one administered on an eye tracker to collect precise eye-movement data and one administered on a touchscreen monitor to collect precise response latencies. We predicted that more gaze transitions and longer response latencies between response options at Time 1 would predict better explicit metamemory monitoring at age 3 (measured as the difference in confidence for accurate compared to inaccurate responses). However, we considered the alternative

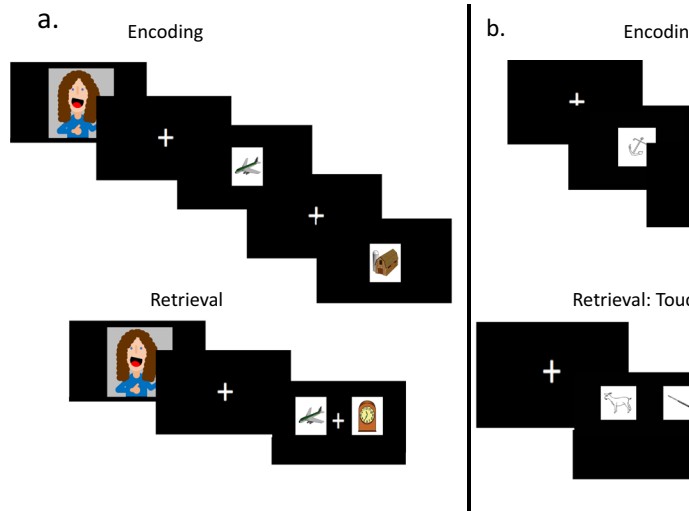

**Fig. 1 | Memory Task Design at Time 1 (a) and Time 2 (b). a** Encoding and retrieval at Time 1 were assessed on two parallel versions of the task, delivered on a touchscreen and on an eye-tracker. During the encoding phase, 2-year-olds saw a video of a female experimenter introducing the colored object images as her own drawings. Then, toddlers saw a series of images presented one at a time at the center of the screen. During the retrieval phase, 2-year-olds saw a video of the same experimenter from encoding inviting them to find her drawings. They then saw pairs of images (1 old and 1 new image) and were asked by the experimenter to either point to (Eye-tracker) or touch (Touchscreen) the image that they had previously seen. Figure has been modified with permission, with any modifications being accurately noted, from Leckey, Bhagath, Johnson, & Ghetti[20], Attention to Novelty Interferes with Toddlers' Emerging Memory Decision-Making. This is an open access article under the terms of the Creative Commons Attribution-NonCommercial-NoDerivs License (https://creativecommons.org/licenses/by-nc-nd/4.0/). © 2023 The Authors. Child Development published by Wiley Periodicals LLC on behalf of Society for Research in Child Development. Changes include changing the colored pictures shown to the toddlers, rearranging the orientation of the encoding and retrieval portions of the task, and exchanging the image of the experimenter for a cartoon image. Stimuli reused from Rossion & Pourtois, Perception (Volume 33, Issue 2) pp. 217-236. Copyright © 2004 by SAGE Publications. Reprinted by Permission of Sage Publications. **b** Encoding and retrieval at Time 2. The encoding phase was the same on both the touchscreen and eye-tracker version of the task. The retrieval phase includes the confidence scale only on the touchscreen version. During the encoding phase, 3-year-olds saw a series of object line drawings presented one at a time. During the retrieval phase, 3-year-olds saw two images (1 old and 1 new image) on the screen and were told to touch the image they had seen before. After they made their selection, three images appeared depicting the levels of decision confidence and were asked to choose how confident they were about their answer. Stimuli reused and reprinted from Journal of Experimental Child Psychology, 65/2, Cycowicz, Friedman, Rothstein, & Snodgrass, Picture Naming by Young Children: Norms for Name Agreement, Familiarity, and Visual Complexity, 171-237, Copyright (1997) with permission from Elsevier.

possibility that response latencies may not become a cue to memory uncertainty until later in childhood[23].

We examined the emergence of metamemory monitoring while assessing changes in memory ability. Memory abilities improve between the ages of 2 and 3 years[28,29] and this change may impact children's capacity to reflect on their memory states because the latter may depend, at least in part, on the quality of the memory signal they ought to reflect on. In accounting for this variable, we were able to test for continuity in memory ability in early childhood. Given that many studies in this age range rely on methods that limit the response demand to assess memory, little is known about how the overt memory abilities at 2 years of age relate to the memory abilities at 3 years of age. If memory abilities at age 2 predict memory abilities at age 3, we would have initial evidence of stability in individual differences in memory.

Finally, we also considered alternative accounts surrounding the factors underlying the emergence of metamemory monitoring. One such factor is the early awareness of mental states. Around 2 years of age, toddlers begin to utilize expressions, such as "I don't know"[30] and gestures, such as shrugging[21], which indicate uncertainty. This early awareness of, and access to, mental states, namely ignorance, may support metamemory monitoring development by giving them the ability to not only identify but also to label their own confidence states[10,31]. Another factor is the acquisition of a theory of mind (ToM), which is the understanding of mental states in oneself and others, and how these mental states may change depending on different situations[32]. ToM develops across early childhood[33], with some researchers attributing early ToM abilities to infants as young as 6 months of age[34]. Furthermore, some researchers argue that ToM

abilities may also promote metamemory monitoring in that general knowledge of the content of the mind and mental states is necessary in order to appreciate ones' own personal mental states[35,36]. Indeed, research has shown that interventions aimed at increasing ToM abilities in preschoolers resulted not only in increases in the preschoolers' ToM, but also in their metamemory abilities[37,38]. Therefore, this understanding of mental states and beliefs, and the knowledge that they may change, may help children gain an understanding of how mental processes are deployed during a task, and how decisions may be either accurate or inaccurate, which would then allow them to subjectively assess their own performance[39]. Thus, additional models tested whether the extent of toddlers' use of mental state language and preschoolers' developing ToM abilities may additionally contribute to metamemory monitoring.

Here, we show that early gaze transitions, response latencies, and memory accuracy at age 2, but not mental state language or ToM, predicts metamemory monitoring abilities at age 3. In addition, we find a longitudinal relation between Time 1 and Time 2 memory accuracy. These findings provide evidence that early behavioral indicators of evidence evaluation and information seeking may be key building blocks for the development of metamemory monitoring observed in older children.

## Results

Descriptive statistics for all variables can be found in Table 1 and correlations between all variables in our longitudinal path models can be found in Supplementary Table 1. We note that at Time 1, memory accuracy was not significantly different from chance across the entire sample in both the eye-tracker ($M = 0.47$, $SD = 0.19$; $t(131) = -1.85$,

$p = 0.067$, $d = 0.16$, 95% confidence interval (CI) = 0.44–0.50) and touchscreen ($M = 0.49$, $SD = 0.18$; $t(143) = -0.71$, $p = 0.482$, $d = 0.06$, CI = 0.46–0.52) versions, but there was a great deal of individual variation which allowed us to examine individual differences in the relation between gaze transitions and accuracy[20]. At Time 2, accuracy scores were similar for the eye-tracker version ($M = 0.61$, $SD = 0.21$) and the touchscreen version ($M = 0.55$, $SD = 0.19$), and they were above chance (i.e., accuracy higher than 0.50) in each version of the task, the eye-tracker version, $t(130) = 6.00$, $p < 0.001$, $d = 0.52$, CI = 0.57–0.64, and the touchscreen version, $t(127) = 2.75$, $p = 0.007$, $d = 0.24$,

CI = 0.51–0.58. Similar to memory accuracy at Time 1, metamemory monitoring (i.e., confidence for accurate trials minus confidence for inaccurate trials) at Time 2 was not statistically significantly above chance at the group level ($M = 0.02$, $SD = 0.36$; $t(122) = 0.50$, $p = 0.619$, $d = 0.04$, CI = −0.05–0.08), but there was substantial individual variation (range = -0.82–1.42) allowing us to identify the longitudinal emergence of this capacity at the pivotal age when children begin to show metamemory monitoring.

### Preliminary analyses

Before conducting our longitudinal path models, we examined whether there were differences in Time 1 variables between the children who completed both Time 1 and Time 2 assessments and the children who only completed Time 1. We found that there were no statistically significant differences in Time 1 age ($t(174) = -1.43$, $p = 0.154$, $d = 0.35$, CI = −1.43–0.23), general vocabulary ($t(174) = -1.34$, $p = 0.182$, $d = 0.33$, CI = −0.21–0.04), response latencies ($t(139) = 0.10$, $p = 0.0918$, $d = 0.03$, CI = −899.51–998.46), gaze transitions ($t(129) = -0.93$, $p = 0.355$, $d = 0.26$, CI = −0.91–0.33), accuracy ($t(163) = -0.73$, $p = 0.469$, $d = 0.18$, CI = −0.10–0.04), or "I don't know" ratings ($t(168) = 0.07$, $p = 0.946$, $d = 0.02$, CI = −0.72–0.77).

### Predicting the emergence of metamemory monitoring

In order to investigate how Time 1 average gaze transitions and average response latencies predicted metamemory monitoring at Time 2, we entered these predictors in a path model along with the same predictors assessed at Time 2 (Fig. 2). We also included mean memory accuracy at both time points, mean confidence at Time 2, Time 1 age, and the difference between Time 1 and Time 2 age, as covariates to ensure that longitudinal relations did not depend on overall differences in the accuracy on the task, overall confidence levels, or variability in age or time to return to the lab. As detailed later, our analysis used all available data with a full information maximum likelihood

**Table 1 | Means and Standard Deviations of Study Variables**

| Time 1 Variables | | | |
|---|---|---|---|
| Variable | N | Mean | SD |
| Age | 176 | 29.00 | 1.73 |
| Gaze Transitions | 131 | 1.95 | 1.10 |
| Response Latencies | 141 | 3786.14 | 1751.03 |
| Accuracy | 165 | 0.48 | 0.15 |
| "I don't Know" Ratings | 170 | 3.59 | 1.51 |
| General Language | 176 | 0.42 | 0.26 |
| **Time 2 Variables** | | | |
| Age | 157 | 41.47 | 3.95 |
| Gaze Transitions | 131 | 1.21 | 0.69 |
| Response Latencies | 124 | 3983.72 | 1551.74 |
| Accuracy | 147 | 0.58 | 0.16 |
| Mental State Language | 142 | 3.69 | 0.77 |
| Theory of Mind | 158 | 0.59 | 0.37 |
| Average Confidence | 128 | 1.49 | 0.45 |
| Metamemory Monitoring | 123 | 0.02 | 0.36 |

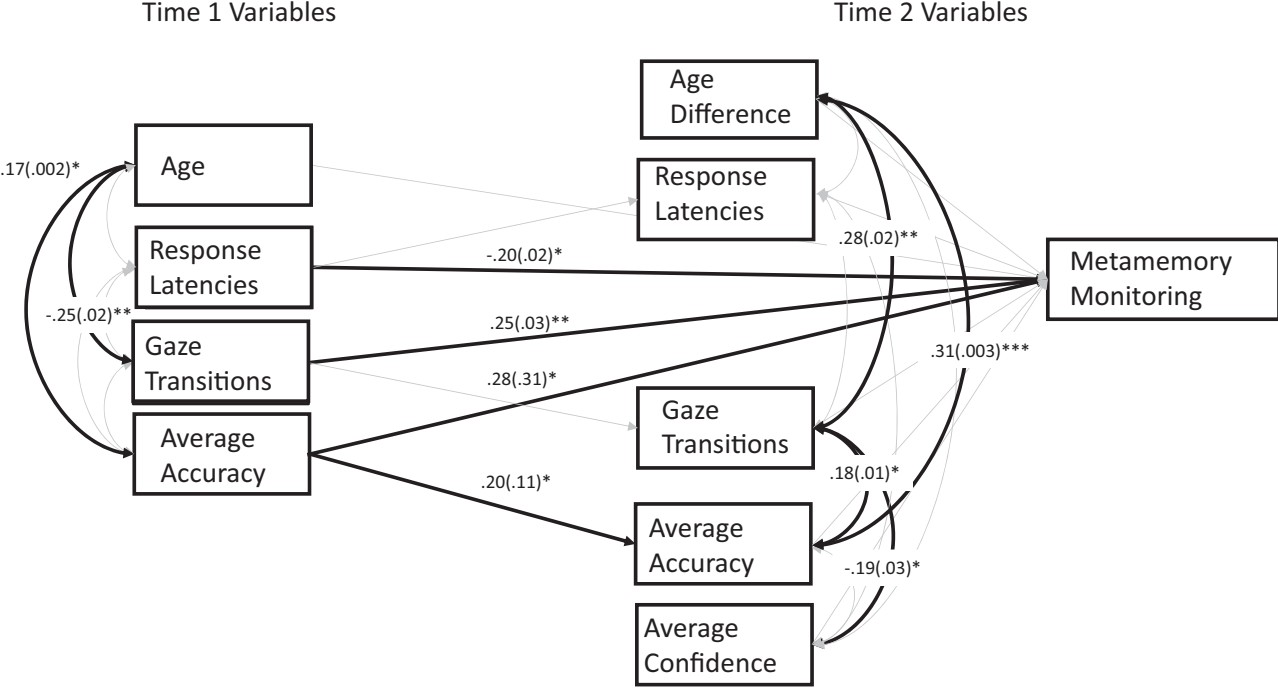

Time 1 Variables Time 2 Variables

**Fig. 2 | Path Model Predicting Metamemory Monitoring at Age 3.** We examined the longitudinal relation between Time 1 variables of age, response latencies, gaze transitions, and average accuracy and Time 2 variables of response latencies, gaze transitions, average accuracy, and metamemory monitoring, controlling for the age difference between time points and Time 2 average confidence. Nonsignificant paths are gray. Standardized betas and standard errors are portrayed for significant paths only ($\beta$(se)). Not pictured: significant covariances between Time 1 average accuracy and age difference, $\beta = -0.17$, SE = 0.003, $p = 0.010$ and between Time 1 response latencies and age difference, $\beta = 0.24$, SE = 0.05, $p = 0.009$. *$p < 0.05$, **$p < 0.01$, ***$p \leq 0.001$.

approach[40] to avoid listwise approaches for dealing with missing data. Our tested model also allowed us to examine the paths from Time 1 variables accounting for Time 2 variables and relevant covariates. Our model was not saturated and absolute and relative model fit indices indicated good fit by the non-significant chi-squared test (robust $X^2(13) = 18.55$, $p = 0.138$), the robust root mean square of approximation (RMSEA) = 0.06, and robust comparative fit index (CFI) = 0.93.

We found significant longitudinal paths from Time 1 gaze transitions to Time 2 metamemory monitoring, $\beta = 0.25$, SE = 0.03, $p = 0.004$, CI = 0.08–0.43, from Time 1 response latencies to Time 2 metamemory monitoring, $\beta = -0.20$, SE = 0.02, $p = 0.016$, CI = −0.37−−0.03, and from Time 1 accuracy to Time 2 metamemory monitoring, $\beta = .28$, SE = 0.31, $p = 0.028$, CI = 0.07–0.49. This indicates that those toddlers who at Time 1 exhibited more gaze transitions between response options during the memory task, responded more quickly, and who showed higher levels of memory accuracy, showed better metamemory monitoring at Time 2. We also found a significant longitudinal path from Time 1 accuracy to Time 2 accuracy, $\beta = 0.20$, SE = 0.11, $p = 0.046$, CI = 0.01 to −0.39. Thus, individual differences in accuracy levels were consistent across the two time points and 2-year-olds who performed better on memory continued to do so one year later. The path from Time 2 gaze transitions and Time 2 metamemory monitoring was not statistically significant ($\beta = 0.03$, SE = 0.04, $p = 0.725$, CI = −0.13 to −0.19; but see Supplementary Note 1 for a multilevel model showing concurrent associations between confidence levels and gaze transitions). However, the association between Time 2 gaze transitions and overall subjective confidence was significant ($\beta = -0.19$, SE = 0.03, $p = 0.044$, CI = −0.37 to −0.01), such that 3-year-olds who switched their gaze more frequently between response options were more likely to report lower overall confidence. This negative association occurred despite a positive association between Time 2 gaze transitions and Time 2 accuracy ($\beta = 0.18$, SE = 0.01, $p = .036$, CI = 0.01–0.35), indicating that 3-year-olds who switched their gaze more frequently were actually being more accurate despite feeling less confident. The association between Time 2 response latencies and Time 2 metamemory monitoring was not statistically significant ($\beta = -0.07$, SE = 0.02, $p = 0.330$, CI = −0.21−0.07). Finally, there were also significant associations between Time 1 age and Time 1 gaze transitions ($\beta = -0.25$, SE = 0.02, $p = 0.007$, CI = −0.42 to −0.08) and Time 1 accuracy ($\beta = 0.17$, SE = 0.002, $p = 0.042$, CI = 0.01–0.33), and between the age difference between Time 1 and Time 2 and Time 2 gaze transitions ($\beta = -0.28$, SE = 0.02, $p = 0.002$, CI = 0.14–0.41) and Time 2 accuracy ($\beta = 0.31$, SE = 0.003, $p < 0.001$, CI = 0.19–0.42).

Thus, our results were consistent with our main hypothesis that early gaze transitions would predict better metamemory monitoring. Moreover, gaze transitions at Time 2 were associated with lower overall confidence, but higher accuracy, suggesting that gaze transitions were helpful to sample memory information, but protracted inspection of the images resulted in lower subjective confidence. In order to determine whether the Time 1 gaze transitions were indeed driven by an ability to assess memory states in service of overt decisions (as opposed to an ability to retain memories), we tested two alternative, exploratory, models in which we replaced: (1) Time 1 memory accuracy with Time 1 novelty preferences from a similar memory task in which children were simply instructed to look at the test probes without making any decisions about them (i.e., Passive Viewing Task; Alternative Model 1) and (2) Time 1 gaze transitions during memory decisions with Time 1 gaze transitions during the Passive Viewing Task (Alternative Model 2).

As a sample, toddlers exhibited a novelty preference that was above chance ($M = 0.56$, $SD = 0.08$; $t(147) = 8.66$, $p < 0.001$, $d = 0.71$, CI = 0.55–0.57), which underscored that two-year-olds retained memory for pictures. Moreover, we observed an average of 2.33 ($SD = 0.71$) transitions between response options in this passive task. For

Alternative Model 1, we kept all paths the same as the original model reported above, only replacing the Time 1 accuracy variable for the Time 1 novelty preference variable. If the development of metamemory monitoring depends primarily on the presence of a memory representation, instead of the ability to report on the memory, then novelty preferences at Time 1 should be expected to predict metamemory monitoring at Time 2. However, if metamemory monitoring depends not just on the mere presence of a memory, but on the ability of the toddler to make overt decisions about their memories, then novelty preferences at Time 1 would not predict metamemory monitoring at Time 2. This model showed good fit (robust $X^2(14) = 15.02$, $p = 0.377$, the robust RMSEA = 0.03, and robust CFI = 0.97). Similar to the first model, we found significant longitudinal paths from Time 1 gaze transitions to Time 2 metamemory monitoring, $\beta = 0.26$, SE = 0.03, $p = 0.005$, CI = 0.07–0.44 and from Time 1 response latencies to Time 2 metamemory monitoring, $\beta = -0.21$, SE = 0.02, $p = 0.015$, CI = −0.38 to −0.04. The path from Time 1 novelty preference to Time 2 metamemory monitoring was not statistically significant, $\beta = -0.03$, SE = 0.44, $p = 0.795$, CI = −0.22 to 0.17. Taken together, the results of the first model and the results of Alternative Model 1 are consistent with the idea that overt memory decisions (as opposed to memory retention alone) is the critical variable predicting later metamemory monitoring.

For Alternative Model 2, we kept all paths the same as Alternative Model 1 and only replaced Time 1 gaze transitions with the Time 1 gaze transitions from our passive task. If the effect of Time 1 gaze transitions is reduced, or even eliminated, with this substitution then it would suggest that being able to suppress novelty preferences during memory decision-making is indeed important for the development of metamemory monitoring. This model again showed good fit (robust $X^2(14) = 12.60$, $p = 0.558$, the robust RMSEA = 0.00, and robust CFI = 1.00). With the gaze transitions from the passive memory task, the path from Time 1 gaze transitions to Time 2 metamemory monitoring was no longer statistically significant, $\beta = 0.02$, SE = 0.05, $p = 0.847$, CI = −0.18–0.22, suggesting that the relation between gaze transitions and metamemory monitoring has to do with toddlers' evaluation of their test probes while engaged in goal-directed memory decisions as tested in the initial model. Therefore, these alternative models underscore the importance of the ability to overtly select a memory response and visual exploration during those decisions for metamemory monitoring.

Next, we tested a model to determine whether frequency of Time 1 "I don't know" use and Time 2 ToM predicted metamemory monitoring at Time 2. To determine this, we added these new predictors into the first model (for pre-registered models with the effects of these variables tested in separate models, see Supplementary Note 2). We also included general language ability at Time 1 as an additional covariate to ensure that longitudinal relations did not depend on overall differences in the variability in general language. Finally, we also included paths predicting Time 2 mental state language and Time 2 ToM from Time 1 gaze transitions and response latencies.

Unlike the previous model, this model had overall poor fit (robust $X^2(27) = 49.25$, $p = 0.006$, robust RMSEA = 0.08, and robust CFI = 0.87). Critically, the paths between Time 1 "I don't know" ratings and Time 2 metamemory monitoring, between Time 2 mental state language and Time 2 metamemory monitoring, and between Time 2 ToM and Time 2 metamemory monitoring were not statistically significant ($\beta = 0.11$, SE = 0.02, $p = 0.202$, CI = −0.06–0.27; $\beta = 0.16$, SE = 0.05, $p = 0.122$, CI = 0.04–0.34; $\beta = -0.16$, SE = 0.09, $p = 0.075$, CI = −0.33–0.01). There was a significant path from Time 1 general language to Time 2 mental state language, $\beta = 0.44$, SE = 0.24, $p < 0.001$, CI = 0.30–0.58, and a significant concurrent path from Time 1 general language to Time 1 use of the "I don't know" expression, $\beta = 0.43$, SE = 0.36, $p < 0.001$, CI = 0.31–0.55. Replicating the results of the first model, we found significant longitudinal paths

from Time 1 gaze transitions to Time 2 metamemory monitoring, $\beta = 0.23$, SE = 0.03, $p = 0.007$, CI = 0.06–0.40, from Time 1 accuracy to Time 2 metamemory monitoring, $\beta = 0.29$, SE = 0.29, $p = 0.014$, CI = 0.10–0.48, and from Time 1 accuracy to Time 2 accuracy, $\beta = 0.20$, SE = 0.11, $p = 0.048$, CI = 0.01–0.39. The path from Time 1 response latencies to Time 2 metamemory monitoring was no longer statistically significant, but followed the same trend as the earlier model, $\beta = -0.17$, SE = 0.02, $p = 0.063$, CI = −0.36–0.02. We also found that the same significant associations between Time 2 gaze transitions and average subjective confidence ($\beta = -0.19$, SE = 0.03, $p = 0.048$, CI = −0.36 to −0.01), and between Time 2 gaze transitions and Time 2 accuracy ($\beta = 0.18$, SE = 0.01, $p = 0.034$, CI = 0.01–0.36). No other significant paths were found, similar to the first tested model. We additionally tested a model in which ToM scores at Time 2 were replaced with scores that were conditionalized by accuracy in the control check question[41] (Supplementary Note 3). Overall fit and results for this model were identical to the one reported above (robust $X^2(27) = 46.75$, $p = 0.011$, robust RMSEA = 0.08, and robust CFI = 0.89).

## Discussion

Metamemory monitoring is fundamental for guiding learning behaviors and decision-making[1,42]. Although this ability does not emerge until between 3 and 4 years of age, recent research has suggested that even infants and toddlers show certain information-seeking behaviors when facing difficult memory decisions, such as turning towards adults for help[8]. These early behaviors have prompted questions about their functional significance for later emerging metamemory monitoring. In this study, we examined whether behaviors of gaze transitions and response latencies predicted metamemory monitoring at 3 years of age.

We found that the extent to which toddlers' gazes transitioned between retrieval test probes prior to committing to a decision at 2 years of age positively predicted metamemory monitoring at age three, consistent with the hypothesis that this behavior provides a foundation for metamemory monitoring one year later. We interpret this finding as suggesting that the tendency to visually explore and compare the response options prior to giving a response, which may indicate information seeking and evaluation, might allow more opportunities for toddlers to recognize differences in their memory states, which in turn might lead to a better ability over time to discriminate between those states as reflected in metamemory assessments. From this perspective, these visual explorations need not be themselves metacognitive (i.e., children are unlikely to have conscious access to their use of visual exploration), but they may reflect information seeking, which enables the emergence of metamemory monitoring over time. Based on prior research[20], this could mean that the development of metamemory monitoring may be supported by attentional control skills in toddlerhood. In a recent paper examining the Time 1 memory data from this current study[20], researchers found that only a subset of toddlers were able to report accurately on their memories when asked, even though all toddlers showed memory for the images based on a novelty preference paradigm. In a follow-up experiment with different participants, researchers utilized an attentional manipulation designed to reduce toddlers' attention towards the novel image which resulted in increased gaze transitions and accuracy[20]. Therefore, individual differences in visual exploration depend, at least in part, on attentional control to inhibit novelty preferences in service of prioritizing goal-directed memory retrieval and associated monitoring operations. Toddlers' continued experience with monitoring their retrieval processes may scaffold metamemory monitoring indexed by confidence judgments differentiating accurate from inaccurate responses.

At Time 2, we found that gaze transitions were associated with overall memory confidence, such that those 3-year-olds who visually explored response options more, were more likely to report lower confidence across all answers. This result suggests that 3-year-olds may have used the extent of their examination of the response options before committing to a response as a cue to overall confidence. In addition, an exploratory analysis revealed that there were more gaze transitions for trials in which 3-year-olds choose "Not so sure" compared to trials in which they chose "Really Sure" (Supplementary Note 1), suggesting that gaze transitions may still inform individual confidence decisions at this age. These differences suggest that visual exploration at age 2 and age 3 may reflect, at least in part, different processes. Whereas at 2 years of age this behavior may primarily indicate the extent to which toddlers inhibit the tendency to examine the novel distracter in favor of their current goals, enabling children to engage in retrieval monitoring, at 3 years of age, when children more readily inhibit such novelty preference, they may additionally interpret the necessity of further visual inspection as an indicator of uncertainty. Intriguingly, individual differences in gaze transitions at 3 years of age are positively associated with concurrent memory accuracy. Thus, although 3-year-olds who engage more in visual exploration end up being more accurate, suggesting that it resulted in more effective decisions, they also felt less confident about their answer. This dissociation between accuracy and confidence is consistent with previously documented dissociations in older children and adults who are more accurate, but less likely to be confident, when the memory task requires more visual comparisons between the two test probes[43–45]. Evidence of such confidence-accuracy dissociation in 3-year-olds underscores the early emergence of heuristic cues used to evaluate memory quality. Future experimental studies should investigate the boundary conditions of early heuristic cue use, including the effect for such variables as test probe similarity and degree of attention distribution.

Although we found a significant correlation between concurrent gaze transitions and memory accuracy at Time 2, we did not find a statistically significant relation at Time 1. We speculate that the strong novelty preferences in 2-year-olds might obscure variability in gaze transitions and explain why we did not find this relation at Time 1. By 3-years of age, children's stronger ability to redirect gaze away from the novel item during memory decisions may better reveal a correlation between gaze transitions and memory accuracy.

We also found that accuracy at Time 1 positively predicted metamemory monitoring as well as accuracy at Time 2. We suggest that toddlers' ability to exert better control on their memory performance, including using memory signals to guide overt decisions, supported toddlers' emerging ability to match their assessment of memory states to confidence reports. Indeed, this has been suggested in prior literature examining how young children may utilize implicit certainty judgements to control their choices on a memory test, even before they can explicitly report on their uncertainty[8,46,47]. Moreover, we tested alternative models in which memory decision accuracy at age 2 was replaced with novelty preference at the same time point and found that novelty preference did not statistically significantly predict metamemory monitoring. The contrast between the finding of the main model and the findings of our alternative models suggests that the availability of a strong memory representation in itself does not support metamemory monitoring and bolsters the idea that the distribution of attentional processes during decision-making and the ability to translate memory signals in decisions may be critical.

With regards to response latencies, we found that this behavior at age 2 predicted later metamemory monitoring, but at age 3 the relation was not statistically significant. However, contrary to our initial prediction, we found that it was shorter, not longer, response latencies at age 2 that predicted better metamemory monitoring. This opposite relation seems to indicate that the relation between response latencies in 2-year-olds and metamemory monitoring in 3-year-olds is not about hesitation. Together with the finding that memory accuracy at Time 1

predicts metamemory monitoring, this finding suggests instead that toddlers' effective memory retrieval is important for future metamemory monitoring. This pattern of results also leads us to speculate that the direct use of response latency as a cue to memory decision confidence emerges later in development. Indeed, this is what has been suggested in prior research[23]. In a study examining whether children between 7 and 11 years of age utilize response latencies as a cue for metamemory monitoring, Koriat and Ackerman[23] found that although all age groups did show an inverse relationship between response latencies and confidence choice, this relation grew stronger with age. Furthermore, task difficulty can also be a factor to whether response latencies are used as a cue during metamemory monitoring[26], with harder questions showing more disparity between age groups for the relation between metamemory monitoring and response latencies. Task difficulty could potentially be a factor in the current study. Although preschoolers' accuracy scores were significantly above chance, the task was still reasonably challenging and far from ceiling. This could impact the extent to which children can use response latencies as a cue, especially if children at this age respond fairly slowly overall, and response latency may capture additional sources of variability (e.g., momentary distraction, dexterity) making this behavior less informative. For this reason, perhaps, children were better at tracking what they did during decisions (i.e., amount of visual inspection of the test probes) than the time that it took them to make their decisions. Additionally, although models linking confidence and response latencies in adults have shown that faster response latencies are linked to lower confidence ratings[27], it is possible that in these young children, there is more noise for these difficult decisions and therefore, they are associating faster response latencies with better, more confident decisions.

Our alternative models tested whether early use of mental state language and ToM may contribute to the emergence of metamemory monitoring, based on theories that suggest connections between these three constructs[48]. It has been suggested that toddlers use of verbal expressions of ignorance, such as "I don't know," may increase their understanding of confidence states[10,31]. Similarly, it has been theorized that the understanding of mental states and how they explain decisions (i.e., ToM) is required before one is able to determine whether a decision is accurate or inaccurate[49,50]. However, neither toddlers' parental reported use of "I don't know" nor preschoolers' ToM ability were statistically significant predictors of metamemory monitoring at age 3. The ToM result is consistent with prior research which showed ToM ability is not related to metamemory monitoring in children 3–8 years of age[51]. Therefore, this suggests that mental state language and ToM may not be essential to the emergence of metamemory monitoring at this early age. These results stand in apparent contradiction to those that reported relations between ToM and metamemory[37,38]. However, in those studies, metamemory was assessed in terms of children's declarative knowledge of how memory works and not in terms of children's ability to monitor their own memory operations. Therefore, the connection between these constructs may be stronger when measures of overt conceptual understanding are involved.

Our results also inform an ongoing debate in the literature surrounding the connections between metamemory monitoring and ToM. Some researchers posit that ToM must precede metamemory monitoring during development because metamemory monitoring would entail redirecting our capacity to understand minds (i.e., engage in mindreading) inwards[35]. Our results are more consistent with proposals that posit that ToM and metamemory monitoring may emerge from separate systems[52]. However, our results do not rule out the possibility that these factors may become more important later in development or matter more in other domains of metacognition. For example, ToM, as indicated by false belief, was associated with a measure of metacognitive control for perceptual decisions in preschoolers[53]. Moreover, prior research examining neural correlates of metacognition in varying domains has suggested at least partially distinct mechanisms[54,55]. Finally, as discussed earlier, ToM seems to be particularly important for the development of metamemory knowledge[37,38].

One potential limitation of this study is that the preschoolers, at the sample level, did not exhibit statistically significant metamemory monitoring abilities. Therefore, we cannot interpret our results as showing that 3-year-olds show this ability at the sample level. This is not surprising, however, given that other studies have shown that metamemory monitoring is not reliable until closer to 4 years of age (e.g.,[5,47,56]). Moreover, individual variability proved to be meaningful (as was the case for individual variability in memory accuracy at Time 1[20]). Nevertheless, future research should investigate these longitudinal relations with a slightly older sample, in order to test whether these early behavioral indicators predict metamemory monitoring even when children possess more robust skills, or whether instead these behavioral indicators are particularly powerful during a developmental transition between implicit behaviors and overt uncertainty monitoring[9].

An additional limitation of this study is that we only assessed metamemory monitoring as the difference in confidence ratings for accurate minus inaccurate trials. Other studies have utilized metacognitive control measures in young children, similar to ones used in non-human animals where participants have the option to skip trials[46]. Future research should test whether the longitudinal relations reported here extend to other measures. More generally, examining multiple outcomes may provide a more comprehensive understanding on the emergence of metamemory.

Overall, we showed that early behavioral indicators of information gathering and evaluation at age 2 predicted metamemory monitoring abilities at age 3, whereas mental state language and ToM did not. These findings not only begin to show convergence between the early metacognitive behaviors seen in infancy and metamemory monitoring in older children but also pave the road for future interventions geared towards young children in order to better prepare them for their entrance to formal education. Future research should examine whether promoting visual inspection of response probes in memory tasks, which may vary in difficulty, supports the emergence of metamemory. This would provide converging mechanistic evidence that visual exploration during a decision is an important precursor to the development of the ability to reflect on one's memory and may pave ways to early intervention.

## Methods

This study, and Leckey et al.[20], were approved by the Institutional Review Board of the University of California, Davis, and informed consent was obtained from all parents. This study was pre-registered and can be found on Open Science Framework at https://osf.io/9wz2m. The Investigators were not blinded to allocation during experiment and outcome assessment. However, experimenters interacting with the children were blind to the specific hypotheses of the study. No statistical method was used to predetermine sample size. We attempted to recruit equal numbers of males and females, but we did not test for differences between sexes, and sex was not considered in study design because we did not have any hypothesis relevant to this variable. Sex was determined by parental reports in our demographics form.

### Participants

A sample of 176 toddlers aged 25–34 months ($M = 29$ months, 93 female) participated in the first assessment (Time 1[20]). The toddlers at Time 1 were the toddlers from Leckey et al.[20], that had been asked to participate in the longitudinal study. Therefore, this current study encompasses a subset of the toddlers reported on in the previous

paper. Time 2 occurred on average 12.41 months (SD = 3.42) later and included 157 children from Time 1, now preschoolers, aged 35–60 months (M = 41.47 months, 84 female). Reasons for participants not returning for Time 2 included our inability to contact them (10), moved out of area (4), and no longer interested in participating (5).

Parents reported their children's race as White (N = 119), Asian (N = 5), African American (N = 5), multi-racial (N = 37), and unreported (N = 10). Parents of 31 children identified them as Hispanic. Families' household incomes were less than $15,000 (N = 5), $15,000–$25,000 (N = 5), $25,000–$40,000 (N = 15), $40,000–$60,000 (N = 29), $60,000–$90,000 (N = 33), more than $90,000 (N = 85) and unreported (N = 4). The children were recruited from a Northern California community, from a database of families contacted from birth records, who had expressed interest in participating in child development studies. None of the children had a history of developmental or speech delays.

### Time 1 Assessments (At 2 years of age)

**Memory Tasks (Fig. 1a[20]).** Toddlers were assessed on two versions of a memory task. Each version had 20 trials, with 2 additional trials in the eye-tracker version that were used as practice trials. Toddlers would first encode 20 images (22 images in the eye-tracker version) and then complete a 20-trial retrieval phase where they saw two images (1 familiar, 1 new), side by side, with the first two trials being practice trials in the eye-tracker version. Toddlers were asked to select the target image by touching it (touchscreen version) or pointing to it (eye-tracker version). The stimuli were drawn from 160 colored line drawings from a widely used database[57] depicting common objects and animals typically known to 2-year-old children. This familiarity was determined by utilizing the MacArthur-Bates Communicative Development Inventories along with age-of-acquisition norms from Morrison and colleagues[58]. Four sets of 40 drawings each were created based on random selection for counterbalancing purposes, and four additional drawings were used as the practice trials before the retrieval phase for the eye-tracker version. Toddlers were assigned to a different set for each version of the task and set use was counterbalanced across participants. Results in two-year-olds from this task have been reported separately[20]. Here, we discuss how we processed and analyzed these data in order to assess their longitudinal relation with those collected in the longitudinal follow-up.

For the eye-tracker version of the task, children sat on their parent's lap, approximately 60 cm from the monitor and the experimenter sat on their left. The stimuli were 10 cm × 10 cm (visual angle 9.53) with 4.45 cm (visual angle 4.24) between them. Parents wore dark sunglasses to ensure that their eyes were not recorded and that they could not view the stimuli. Parents were asked to hold their toddler to prevent any excessive movement or leaning forward. They were also asked to not speak to or engage with their toddler during the task. Before each phase, toddlers underwent standard infant calibration procedures[20]. Each phase proceeded when toddlers' gaze was captured at all five calibration points on the screen. Default Tobii fixation filter settings were used for data reduction (velocity threshold: 35 pixels per sample; distance threshold: 35 pixels).

The encoding phase was introduced to the toddlers by the experimenter purportedly showing them a friend's drawings: "Now my friend Julia is going to show us some of her drawings." Then, toddlers saw a video of a female experimenter presented at the center of the eye-tracker screen introducing the pictures as her drawings. After, toddlers viewed the 22 pictures one after another. Each picture was presented individually at the center of the screen for 3 seconds and a white fixation cross, paired with a "ding" sound, was shown in between pictures to maintain the toddlers' attention to the center of the screen. Toddlers were never overtly instructed to remember these images.

After the second calibration procedure, toddler completed the retrieval phase. The experimenter first introduced the task and

instructed the toddlers to help them find the confederate's drawings by saying, "Now we are going to help Julia find her drawings" and they were told to indicate their answer by pointing. Two practice trials followed to verify that the toddlers responded to the instructions by pointing. Before each trial, the experimenter asked the toddler to indicate the previously seen picture (i.e., "Which picture did Julia show you before? Only one is Julia's!") on a screen with a fixation cross and then pressed the space bar to present the trial. If the toddlers pointed to the correct picture, they were told that they were correct, and if they pointed to the wrong picture, they were corrected and reminded that they needed to point to the old picture. The 20 test trials followed the same procedure as the practice trials, except that no feedback was provided. As soon as the toddlers chose a picture, the experimenter keyed in the pointing response which brought up the next screen with a fixation cross. If toddlers refused to choose a picture, the experimenter pressed the space bar button and that trial was removed from analyses.

The touchscreen version of task was administered on a 17-inch Planar PT1701MU LCD touchscreen monitor with 1280 × 1024 resolution. The stimuli were the same size and distance apart as the eye-tracker task. This task was identical to the eye-tracker task except that the child sat alone about 60 cm in front of the touchscreen monitor on a child-sized chair, for each retrieval trial the toddler was instructed to respond by touching the screen, and there were no practice trials. As soon as the child touched a side of the screen, the task advanced to a blank screen before starting the next trial. If a child refused to respond, the experimenter keyed in a separate code to remove that trial from analysis and moved on to the next trial.

The variables obtained from the memory tasks included gaze transitions, response latencies, and accuracy. Gaze transitions were obtained from the eye-tracker version of the task. In order to calculate gaze transitions, we used Tobii Studio software to create areas of interest (AOIs). These AOIs encompassed individual square images surrounding the target and distractor stimuli respectively, so that each trial had a target AOI and a distractor AOI. Our gaze transitions variable was calculated by counting how many times the children's gaze transitioned from one stimulus to the other during a trial. We defined a switch as a fixation on an AOI that was preceded by a fixation to the other AOI, including instances in which there were fixations on other (non-AOI) areas of the screen in between fixations to AOIs. The first fixation to an AOI in a trial was not counted as a switch, so the minimum number of transitions in a trial was zero. The final gaze switch variable was an average of gaze transitions from all valid trials. Response latencies were obtained from the touchscreen version of the task and were measured as the time from when the stimulus was shown on the screen to the time when the participant touched the screen for their response. The final response latency variable was an average of response latencies from all valid trials. The accuracy variable was an average of the accuracy scores from the touchscreen and eye-tracker versions of the task. To calculate this average, we first calculated the participants average accuracy for both tasks separately, and then took an average of those two scores. For the participants who did not complete both versions of the task we predicted missing values by utilizing a linear regression model predicting either touchscreen accuracy from eye-tracker accuracy or vice versa. This decision to impute missing accuracy scores from the single available accuracy score was not explicitly stated in our pre-registration. However, we deemed it important to adopt current approaches[59] and avoid pairwise deletion. The resulting accuracy average was thus obtained from two scores across all participants.

Not all participants contributed data in both tasks due to some toddlers being uncooperative in one (33) or both tasks (6), not returning for all sessions (3), computer malfunction (8), they completed an advanced pilot version of the memory task (9), or a

combination of these reasons (2). Therefore, 133 toddlers contributed eye movements data and 145 toddlers contributed response latency data.

Before calculating the accuracy and gaze switch variables for the eye-tracker version, we first eliminated trials for which children did not provide an answer. This resulted in 57 trials (2.12%) across 18 toddlers being eliminated from analyses. Next, we removed retrieval trials for which the toddlers had not looked at the picture during the encoding phase, indicated by no fixations during the 3-s period of encoding phase. This criterion resulted in the exclusion of 155 trials (5.77%) across 68 toddlers. Finally, we also removed trials for which the eye-tracker did not measure any look time to either AOI, target or distractor during the retrieval phase. This criterion resulted in the exclusion of 415 trials (15.45%) across 80 toddlers being eliminated from analyses. Once these eliminations were made, accuracy and average gaze transitions for each participant were calculated. Additionally, after average gaze transitions were calculated, the distribution of the variable was examined and several potential outliers were visually seen. Therefore, we removed any average gaze transitions that were ±3 standard deviations from the mean. This resulted in 1 average gaze transition score being removed.

Before calculating the accuracy and response latency variables from the touchscreen version, similar to the eye-tracker version, we removed trials for which children did not provide an answer. This resulted in 10 trials (.36%) across 6 toddlers being eliminated. Next, we removed any trials with response latencies less than 700 ms in duration. These trials were likely to be responses produced before processing the stimuli or trials in which the toddler was inattentive. We followed previous research in toddlers which has used 700 ms as a touching response latency response cutoff[21]. This criterion resulted in 20 trials (.72%) across 14 toddlers being eliminated from analyses. Finally, we also removed trials where the z-scored response latencies across each individual participant were ±3 standard deviations. This resulted in 71 trials (2.56%) across 71 toddlers being eliminated from analyses. Once these eliminations were made, average response latencies and accuracy for each participant were calculated. After average response latencies were calculated, three scores were removed for being ±3 standard deviations away from the mean.

**Passive viewing task.** The passive viewing task was an additional memory measure that was utilized in two alternative, exploratory path models, which did not require toddlers to make any memory decision as in the memory task described above. This task was delivered on the eye-tracker and was identical to the main memory task above except that during retrieval, toddlers were simply encouraged to look at the pictures during the retrieval trials ("Hey look!) as opposed to being asked to select the familiar picture and each retrieval trial was presented for 10 seconds for the toddlers' inspection.

We calculated novelty preference from looking behaviors, given the large body of work that shows that this is an informative index of memory retention[60] unencumbered by decision processes. Specifically, we used Tobii Studio software to create AOIs around the target and distractor. We then calculated novelty preference for each trial by dividing the amount of time the toddler spent looking at the distractor AOI during the entire trial by the amount of time spent looking at either the target or distractor AOIs during the trial. The final novelty preference variable was an average of novelty preferences from all valid trials.

Not all participants at Time 1 contributed data in this passive task due to some toddlers being uncooperative (9), computer malfunction (9), or because they completed an advanced pilot version of the memory task (10). Therefore, 148 toddlers contributed data. Before calculating the novelty preference variable, we first eliminated recognition trials for which the toddlers had not looked at the picture during the encoding phase, indicated by no fixations during the 3 second

period of encoding phase. This criterion resulted in the exclusion of 320 trials (9.61%) across 87 toddlers. We also removed trials for which the eye-tracker did not measure any look time to either AOI, target or distractor during the recognition phase. This criterion resulted in the exclusion of 497 trials (14.92%) across 88 toddlers being eliminated from analyses. Once these eliminations were made, novelty preferences for each participant were calculated by taking total time looking at the novel image divided by the total time looking at both the novel and the old image. We also calculated the number of transitions between the two response options as was done in the memory task included in the main manuscript and the final gaze transition variable was an average across all valid trials.

**Use of ignorance expressions.** Time 1 use of ignorance expressions was assessed via parental report. During the laboratory visit we had parents report how frequently their toddler used the phrase "I don't know" in typical, everyday conversations on a five-point scale ranging from 0 (never/not yet) to 4 (often).

**General vocabulary.** Time 1 general vocabulary was assessed with MacArthur-Bates Communicative Development Inventory-II language questionnaire[61]. This is a questionnaire filled out by the parent which asks about their toddlers' vocabulary, grammar, semantics, pragmatics, and comprehension. For this study, we used the toddlers' vocabulary score, resulting from a parent identifying and marking which of 100 words their toddler could express. The vocabulary score was calculated by summing the number of words identified by the parent and creating a proportion score by dividing by the total number of known words by the total number of words on the list (i.e., 100). This variable was not originally listed on our pre-registration. However, we deemed it important to account for it when assessing the relation between children's mental state language and metamemory monitoring.

### Time 1 procedure

Toddlers participated in 3 sessions spaced about a week apart and after each session they received a book for their participation. Before data collection for each session, the experimenter played with the child outside of the testing room for about 5 minutes in order to build rapport and increase comfort. During Session 1, parents completed demographic and vocabulary questionnaires and reported on their toddler's use of ignorance language. Additionally, the toddlers completed the passive viewing memory task. During Session 2, the eye-tracker version of the memory task was administered and during Session 3, the touchscreen version of the memory task was administered. The touchscreen task was always administered in the third session because pilot testing revealed that toddlers were more likely to reach forward to touch the eye-tracker if they underwent the touchscreen task first which interfered with eye movement data collection.

### Time 2 Assessments (At 3 years of age)

**Memory Tasks (Fig. 1b).** At Time 2, preschoolers completed a memory task that was similar to the task used at Time 1 and to previous studies with this age group[5]. Preschoolers were additionally asked to provide confidence judgments (touchscreen version only). Stimuli included 104 line drawings of objects and animals[62]. Eight images were used for practice trials in the touchscreen version, 20 images were used for confidence practice in the touchscreen version, and 80 images were used for counterbalancing of the retrieval test pairs for both versions.

The touchscreen version began with the encoding phase and utilized the same setup as Time 1. The experimenter introduced this phase as the picture game and preschoolers were told that they were going to see some drawings on the computer and their job was to touch the picture as soon as they saw it. This was to ensure that the

children attended to the images. The experimenter proceeded to show them how to touch the screen on the first two pictures to ensure that the child understood the task. Pictures were shown on the screen for 2 seconds each and in between each image there was a screen with a fixation cross paired with a "ding" sound. Preschoolers were not overtly instructed to remember these images.

The encoding phase was followed by retrieval training. In order to familiarize them with the task demands, preschoolers completed 4 practice trials. For each trial they saw a pair of images on the screen, one studied image and one new image, and were told to select the image, by touching, that they had seen during the picture game. For each of the practice trials, they received feedback about their accuracy. If they were correct, the experimenter would give positive feedback, (e.g., "Great job! We chose this drawing because we remembered that we saw the seahorse before!") and when they were incorrect, the experimenter would correct them (e.g., "Oops, we actually saw the seahorse before. But that's okay because it's just practice.").

After the retrieval practice, the experimenter would train the preschoolers on using the confidence scale. The confidence scale was a 3-point pictorial scale with pictures of a child displaying facial and body expressions of low, moderate, or high confidence for each point of the scale. The same child image was used for each level and could be interpreted as being either female or male. The experimenter first verbally described each level of the monitoring scale. Low-confidence was described as being "not so sure about your answer", moderate confidence as "kind of sure about your answer", and high confidence as "really sure about your answer". After, the preschoolers completed 10 practice trials where they were asked to find the picture that they had seen previously during the picture game and then reported their confidence for that answer by selecting one of the confidence scale's images. The experimenter would provide feedback on their confidence scale selection; giving positive feedback when the selection seemed to match the outward expression of uncertainty (e.g., "Great job! You were really sure about that one, so you touched this face."), and prompting them to touch another image when their selection did not match the outward expression of uncertainty (e.g., "Hmm, it seemed like you were not so sure about that one. Remember, if you're not so sure about your answer, this is the face you touch."). After the 10 practice trials, the experimenter verified the preschoolers understanding of each point on the confidence scale and stressed the importance of using all three pictures during the rest of the task. Preschoolers then completed 20 test trials, where no feedback was given. They were asked to first identify which picture they had seen during the picture game then immediately after making their choice, they reported their confidence in their choice. Reminders of the usage of the scale were given if the preschooler chose the same scale response for 4 trials in a row (e.g., "Remember, you should use all of the faces to tell me about how you really feel! Because sometimes, you're really sure, sometimes you're kind of sure, and sometimes you're not so sure, and those are all great answers!"). If the preschoolers refused to submit an answer, the experimenter would press a separate key and that trial was taken out of analysis.

The eye-tracking version was identical to the touchscreen version. However, no training on the task was done, they were not asked to report their confidence levels, and they encoded a new set of images. In addition, to ensure less movement for accurate eye movement data, the preschoolers were given a soft-tipped "wand" in order to point to their choice, and the experimenter would key in their response. As in the touchscreen version, for any trial where the preschooler refused to submit an answer, the experimenter would press a separate key, and that trial was taken out of analysis.

Similar to Time 1, response latencies and accuracy were taken from the touchscreen version, gaze transitions and accuracy were

taken from the eye-tracker version, and data processing and calculations were the same for all variables. Two additional variables were taken from the touchscreen task, namely average confidence and metamemory monitoring. Confidence responses for each trial were coded as 0 ("Not so Sure"), 1 ("Kind of Sure"), or 2 ("Really Sure"). The average confidence variable was then calculated by taking the average of the confidence for all valid trials and could take on values between 0 and 2. The metamemory monitoring variable was calculated by first averaging the confidence ratings for the accurate and inaccurate trials separately and then subtracting the average confidence for the inaccurate trials from the average confidence for the accurate responses. Not all participants at Time 2 contributed data in both tasks due to being uncooperative in one (15) or both tasks (3), computer issues (9), not returning for all sessions (3), ran out of time during the session (2), or they completed an advanced pilot version of the touchscreen task (4). Therefore, 134 preschoolers contributed eye movements data, and 128 preschoolers contributed response latency and confidence data.

Similar to Time 1, trials were removed before analysis for not providing answers, quick response latencies, and no looking time measured to stimuli. For response latencies, 3 trials (0.12%) across 3 participants were removed due to preschoolers not providing a confidence rating, and no trials were eliminated due to preschoolers not providing an answer. Nine trials (.36%) across 7 participants were removed for responses being below the 700 ms cutoff. Sixty-three trials (2.50%) across 63 preschoolers were eliminated due to our outlier cutoff of 3 SD above or below the individuals mean. Once average response latencies were calculated, four scores were removed due to being ±3 standard deviations away from the mean. For gaze transitions, there were 74 trials (2.76%) across 22 preschoolers which were eliminated due to preschoolers not providing an answer. There were 399 trials (14.86%) across 98 participants removed due to preschoolers not looking at the image during encoding. Finally, there were 304 trials (11.32%) across 83 preschoolers eliminated from analysis due to no looking time measured towards the target or the distractor. No average scores were removed for being outliers of ±3 standard deviations away from the mean.

**Mental state language.** Time 2 mental state language was assessed with an adapted version of the Cognition portion of the Internal States Language Questionnaire[63]. Parents were asked to report on their child's use of mental state language sentences, which included "I think", "I know", "I guess", "I'm sure" and "I don't know". Parents rated how often their child used each of these phrases on the same five-point scale used at Time 1 from 1 ("never/not yet") to 5 ("often"). The mental state language score was the average across the 5 ratings.

**Theory of mind.** Time 2 theory of mind was assessed using two versions of the explicit false belief task from Wellman and Liu[32]. Three-year-olds completed this task on the same touchscreen monitor as the memory task. Preschoolers were introduced to a character looking for their lost item (e.g., mittens), which was hidden in one of two locations. The preschooler was told that the character's item was really in one of the locations (e.g., in a backpack) but the character thought the item was in the second location (e.g., in the closet). After hearing the story, children were asked where the character would search for the item. If the child answered correctly (chose the place where the character thought the object was) they were awarded a score of 1. If they answered incorrectly (chose the place the object actually was) they were awarded a score of 0. The scores from both versions of the task were averaged for a final theory of mind score which could take on values of 0, 0.5, or 1. Additionally, after the child was asked where the character would search for the item, they were also asked where the item was really located, as a control check question (Version 1: $M = 0.43$, $SD = 0.50$; Version 2: $M = 0.59$, $SD = 0.49$).

## Time 2 procedure

Preschoolers participated in 3 sessions spaced about a week apart and they received a book after each session for their participation. Before data collection for each session, the experimenter played with the child outside of the testing room for about 5 min in order to build rapport and increase comfort with the laboratory environment. During session 1, preschoolers completed one version of the false belief task. During session 2, preschoolers completed both versions of the memory task (eye-tracker and touchscreen) and parents completed the mental state language questionnaire. During session 3, preschoolers completed the second version of the false belief task. Additional assessments were also completed during the three sessions which fall beyond the scope of the present research.

## Analytical approach

All t-tests reported here are two-tailed tests and all t-tests include Cohen's d as an estimate of effect size. To test the main hypotheses, we utilized path models and examined both concurrent and longitudinal relations between Time 1 and Time 2 behaviors and Time 2 metamemory monitoring, while accounting for Time 1 age, the difference in age between Time 1 and Time 2, accuracy across tasks, and Time 2 levels of average confidence, gaze switching, response latencies, and mental state language. We tested our models using the RStudio package lavaan, which uses full information maximum likelihood to utilize all available data while accounting for patterns of missing data[40]. Data was found to meet assumptions of path models, including no multicollinearity between variables, relationships between variables were linear, and residuals and the main dependent variable (the confidence differential) followed a normal distribution. All path estimates come from the "std.all" option in lavaan which uses all path information to determine the standard estimates for the paths. These standardized estimates for the paths can be interpreted as effect sizes. We determined fit of our models by a non-significant chi-square difference test, a comparative fit index of 0.9 or greater, and a root mean square error of approximation of 0.06 or less. Finally, response latencies were transformed from milliseconds to seconds and age variables were transformed from months to years in order to reduce the order of magnitude of standard errors to be similar to the other predictors in our models.

## Reporting summary

Further information on research design is available in the Nature Portfolio Reporting Summary linked to this article.

# Data availability

The raw datasets generated and analyzed during the current study are available in the Open Science Framework repository, https://doi.org/10.17605/OSF.IO/MZ3XJ[64]. Datasets from referenced Time 1 data[20] are available in the Open Science Framework repository, https://osf.io/sr9fq/.

# Code availability

The code generated and used during the current studies are available in the Open Science Framework repository, https://doi.org/10.17605/OSF.IO/MZ3XJ[64].

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

## Acknowledgements

This work was supported by a grant from the National Science Foundation (NSF; BCS1424058) to S.G. Any opinions, findings, and conclusions or recommendations expressed in this manuscript are those of the authors and do not necessarily reflect the views of the NSF. The funders had no role in study design, data collection and analysis, decision to publish or preparation of the manuscript. S.L. was supported by National Institute of Child Health and Human Development Grant F31HD102153.

## Author contributions

S.G. developed the study concept. S.G. and S.L. finalized the study design. S.L. performed data collection. S.L. and D.S. contributed to data analysis and interpretation under the supervision of S.G. All authors drafted the manuscript, provided critical revisions, and approved the final version of the manuscript for submission.

## Competing interests

The authors declare no competing interests.
