## [Transparent Peer Review file · Nature Communications]

Two-year-olds' Visual Exploration of Response Options during Memory Decisions predicts Metamemory Monitoring One Year Later

Corresponding Author: Dr Sarah Leckey

Version 0:

Reviewer comments:

Reviewer #1

(Remarks to the Author)

This is a very nice study testing the relationship between behavioral manifestations of uncertainty monitoring during early childhood and later emerging abilities to provide metacognitive reports about memory accuracy. The findings are really interesting in that a longitudinal relationship is found between early information seeking / uncertainty monitoring and the later emerging ability to report about the accuracy of one's own memories on a (picture-based) confidence scale. It is also remarkable that gaze transitions were associated with confidence judgements at Time 2, a finding that parallels findings with adults but is still important because it confirms previous studies suggesting that gaze transitions might reflect confidence in early childhood. Overall, I applaud the authors for this really nice contribution to the literature.

Major:

To me the beginning of the abstract does not reflect the state of the art nor the aim of the study appropriately... It is also not very clear (e.g., how is uncertainty monitoring not a form of introspection)? I would suggest rewriting the first two sentences at least; what we don't know is how early behavioral manifestations of uncertainty monitoring (hereafter UM) relate to the development of more complex and abstract forms of metamemory reflected in reports (hereafter MR), and this is what the study tries (and manages) to test; at the moment it reads as if little was known about memory monitoring during early childhood at all, which is not correct.

It would be nice to either define precisely and/or avoid using the term "introspection" (cf also my remark above): this is a rather ambiguous term.

The introduction lacks a description of computational models linking confidence and RTs in adults (e.g., see Pleskac & Busemeyer for an example). Many studies have been dedicated to this issue in adults, and could inform the fuzzy findings obtained in children (e.g., it is possible that the speed-accuracy tradeoff that governs the relationship between confidence and RTs is noisier for decisions that are more difficult for children); a similar point could be made for the discussion.

It would be good to specify briefly what the main measures are more at the beginning of the results section (e.g., what is metacognitive monitoring exactly). Similarly, the control task using novelty preferences is not presented and appears suddenly in the results section, this could be highlighted more at the stage of the hypotheses because it is a nice control?

The contrast between the lack of findings relating ToM and language measures to MR as compared to the UM measures is striking! This has important theoretical implications and supports dual models of metacognition over mindreading models of metacognition, this could be emphasized more in the discussion (the theories are briefly mentioned in the introduction but the findings are not contextualized within this theoretical debate in the discussion).

Minor:

l.95: if I understand correctly this study concerns a subpart of the current dataset? If so it would be good to specify this here.

l.239: I don't understand "despite being more accurate" here? I thought this result means that there is an association between gaze shifts and overall subjective confidence regardless of type1 and type2 accuracy (since this is entered as a covariate in the model)? This part of the results section is unclear...

l.292-294 : "may indicate information seeking and evaluation, might allow more opportunities for toddlers to recognize differences in their memory states" and "From this perspective, these visual explorations need not to be themselves metacognitive" : these two sentences are contradictory ! I think you mean that it is not metamemory yet? Or perhaps you mean these are not metacognitive judgements/reports? But if there are evaluating and seeking information they are doing something metacognitive... this wording is not really consistent with the reported findings that there is some specific longitudinal continuity.

Some recent articles are extremely relevant but not covered in the introduction (circa l.71-80) – as argued by the authors, the longitudinal relationship between early UM and MR remained to be measured, but there were other indications already in the literature that the early manifestations of UM are metacognitive in nature rather than reflecting simpler mechanisms of risk avoidance (see for instance: Bazhydai et al., 2020; Dautriche et al., 2022; Lucca et al., 2020; Kuzyk et al., 2019).

Fig.3 caption lacks details (e.g., specifying what how betas and SE are depicted in the figure). It might be a good idea to use thicker lines for significant paths and lighter ones for non-sig paths, the figure is a little bit difficult to read at the moment...

Reviewer #2

(Remarks to the Author)

This paper reports results from a longitudinal study examining the emergence of metamemory in toddlers. At time point 1, a large sample of two-year-olds completed a memory experiment. At time point 2, the now three-year-olds returned to complete an additional memory experiment and test of metamemory monitoring. The authors found that behaviors at time 1 predicted behaviors at time 2. Specifically, participants who had higher memory accuracy, faster response latencies, and more gaze transitions between response options at time 1 demonstrated better metamemory at time 2. The authors suggest that these results indicate that these 'information-seeking' behaviors at time 1 (i.e., gaze transitions) may be precursors to the emergence of metamemory.

This study has several notable strengths. I thought the question about the precursors to the developmental emergence of metamemory ability was very interesting and well-motivated, and the authors have collected an impressive dataset including a large number of participants at two timepoints. However, I have serious concerns about the robustness and replicability of the results, which I outline in more detail below. I also have concerns about the interpretation of the findings.

1. The authors state the study was preregistered, but the preregistration was created after (or on the same day) that the study was submitted, and (as noted in the preregistration) the data were already analyzed for conference abstracts prior to the preregistration. Thus, I don't believe the authors should describe these analyses as "preregistered" as the "hypotheses" stated in the preregistration were determined after the data had already been analyzed. At the very least, the authors should note in the manuscript that the preregistration took place after data analysis. The primary reason I am concerned about the preregistration occurring after analyses is that the authors have a rich dataset with many measures, and it seems as though they have not appropriately corrected for multiple comparisons. Most reported p values are between .01 and .05, which suggests that the effects they do report may be small and unreliable. The authors have previously used other measures of memory and metamemory (e.g., novelty preferences, timecourse of novelty preference, parameters estimated from drift diffusion models) and so it is not clear why they chose the specific measures they did (i.e., gaze transitions and response latencies) or whether they tested other measures as well. I realize that asking the authors to replicate their findings in an additional sample is unreasonable given the fact that this data is longitudinal. But, particularly given the lack of evidence for significant metamemory monitoring in three-year-olds as a group, I'm worried that some of the reported relations may be noise.

2. Related to my last point, the average metamemory score at time 2 is close to 0. Thus, it's not clear how many three-year-olds actually demonstrate above-chance metamemory. This is particularly concerning given the lack of the relation between gaze transitions at age 3 and metamemory at age 3, as well as the lack of the relation between response latencies at age 3 and metamemory at age 3. I thought the authors' speculative interpretation of the absence of the gaze transition effect was interesting and thoughtful, but not convincing without more data supporting it. It would be helpful for the authors to show a distribution of these metamemory scores, as well as to examine other potential behavioral differences between accurate and inaccurate trials to see if they differ more strongly in participants with stronger metamemory. For example, my understanding is that the response latencies measure is a mean across all valid trials. Could the authors compute participants' response latencies separately for inaccurate vs. accurate trials? Perhaps participants who show a greater influence of accuracy on latency may also show stronger metamemory. Similarly, perhaps gaze transitions also differ for participants with better metamemory across accurate vs. inaccurate trials. If these relationships exist, it would help support the idea that some participants are in fact showing metamemory at age 3.

3. I think the idea that visual exploration at time 1 predicts metamemory at time 2 is very interesting, but I found the gaze transition variable hard to interpret, particularly because the relation did not hold when looking only at the time 2 data. In prior papers, the authors have reported that gaze transitions or more distributed patterns of attention are related to memory

accuracy. They found that all toddlers demonstrated an initial novelty preference, but toddlers who were able to inhibit this novelty preference and focus on the goal of identifying the old image (i.e., make a gaze transition to the old image) made more accurate memory responses. So it seems like all three time point 1 predictors (higher memory accuracy, faster response latencies, more gaze transitions) are associated with better memory performance, and that better memory performance at time 1 may predict metamemory at time 2. It is not clear to me that any of these measures are necessarily direct precursors to metamemory ability. It would be helpful for the authors to report whether gaze transitions and latencies at time 1 predicted time 2 memory accuracy. It was not clear to me if these relations were tested, but if they do, then this may suggest that they are stronger indices of memory ability (vs. metamemory) and that memory ability itself predicts the emergence of metamemory.

More minor points:

1. I'm a little confused about the prior publication of the Timepoint 1 data. The paper that has been published with the Timepoint 1 data (I think) has 206 participants (perhaps I am looking at the wrong Leckey et al., 2023?). Does this study include only a subset of them?
2. It is somewhat difficult to interpret Figure 3 (because it's not always clear what labels correspond to what lines) but it looks as though there are strong effects of age that the authors should report in the main text.

Reviewer #3

(Remarks to the Author)

The current longitudinal study examined the development of toddler and preschool children's early memory and indicators of metacognition to predict later memory and decision confidence. Findings showed that early gaze transitions between responses, shorter response latencies, and memory accuracy at Time 1 predicted metamemory monitoring at Time 2. Alternate possibilities such as novelty preference and the role of mental state language and theory of mind were examined but ruled out as contributing to the current findings. Findings illuminate some of the early building blocks of emerging memory introspection.

This study presents important findings for the memory and metacognition fields. The longitudinal evidence is powerful and allows for testing the emergence of these abilities and examine their predictive power for later emerging abilities. The study has been rigorously conducted and methods are described in sufficient detail (although see some comments below), the analyses are appropriate, and the results are interpreted accurately. I especially appreciate that the authors took the care to rule out some alternative possibilities surrounding the role of novelty preference as well as mental state language and theory of mind. Given the strength of this study and manuscript, I just have relatively minor comments that are aimed at improving clarity and readability of the manuscript.

Comments:

1. On page 3, line 64: I wasn't clear what "asking for a cue" meant after an inaccurate response. Can this be clarified or expanded to make the meaning clearer?
2. In the description of Leckey and colleagues (2023) on page 5, it was not at first clear to me that looking preferences were also measured in the active condition, until I read the sentence about the high performing group on line 107. Maybe it's worth saying this explicitly?
3. I suggest that the author put the instructions to children "Now we are going to help Julia find her drawings" and "Which picture did Julia show you before? Only one is Julia's" closer together in the text, as I wondered at first whether children only received the first part of the instruction which seems quite vague without the follow-up question before each trial.
4. I found the phrase "The vocabulary score was calculated by taking the average of the 100 words that the parents checked off" a bit confusing. Did both parents complete the MacArthur-Bates Inventory and so you averaged the two parents scores? Or was the score just the number of words that one parent had selected/checked off. The reference to "average" threw me off here.
5. On page 26, line 581: did the authors mean that children were told that they did not see the seahorse before? If children were incorrect, it seems like they should have been told that they did not see the seahorse previously.
6. It would be nice to include the 3-point pictorial confidence scale as a figure, or perhaps in supplemental materials.
7. Were children asked control or comprehension check questions on the explicit false belief task (e.g., where are the mittens really?). If so, could accuracy on these questions be reported.
8. Please ensure Results section is written in past tense (e.g., "metamemory monitoring at Time 2 is not..." – should be "was not"; "but there is substantial individual variation" should be "there was substantial...").

Version 1:

Reviewer comments:

Reviewer #1

(Remarks to the Author)

The authors have successfully addressed all of my comments.

Reviewer #2

(Remarks to the Author)

In my original review, I noted the interesting question the study addresses but raised concerns about its replicability. The authors have now assuaged some of my concerns, though I believe some further edits to the text are needed to clarify which analyses were preregistered.

First, I apologize for mistakenly interpreting the date of the preregistration. I now see that the preregistration was submitted well in advance of the manuscript's submission. It does appear as though the primary path model of interest was preregistered (though again, the preregistration was submitted after preliminary analyses were conducted, so I remain skeptical of its utility. I will take the authors' word that they did not conduct analyses of the longitudinal data until after preregistration).

That said, some of the analyses reported in the manuscript seem to deviate from those that were preregistered, and the authors should note these in the text. The preregistration specifies separate analyses for mental state language and theory of mind, but those these appear to be combined in the manuscript. In addition, the manuscript models include more variables than those specified in the preregistration. This is relatively minor as these results were not the primary ones of interest in the manuscript, but the authors should note this deviation. It is also not clear why the researchers chose to impute missing accuracy scores from participants who did not complete both versions of the task (an analytic decision not specified in the preregistration).

I am still left wishing that the authors could replicate their complex (and small) effects in an additional sample. I recognize this is not a reasonable request given the scope of data collection, and so I don't believe this concern should prevent publication of the manuscript, particularly given the positive comments from the other reviewers. But I would encourage the authors to split their data into discovery and replication samples in the future.

Reviewer #3

(Remarks to the Author)

I am satisfied with the revisions that the authors have made in response to my concerns and the other reviewers' concerns. I recommend accepting this manuscript.

Reviewer #4

(Remarks to the Author)

Manuscript ID: NCOMMS-23-61769A-Z

Manuscript Title: Two-year-olds' Behavioral Indices of Memory Evaluation Predict the Emergence of Metamemory Monitoring One Year Later

Summary: While I did not serve reviewer on the initial submission, the current manuscript is a revision addressing a set of concerns from the initial reviewers. I first reviewed the manuscript, then viewed the original set of concerns and rebuttal letter. The manuscript addresses a novel and interesting topic of identifying behavioral precedents at age 2 to emerging metamemory monitoring at age 3. The authors have thoughtfully addressed this question, using longitudinal data and path modeling to simultaneously assess questions of interest. Further, they consider plausible alternatives and give space for these alternative theories to play a role in other aspects of cognitive development. Overall, I am impressed by the authors' work. I think the authors have addressed the prior set of reviews quite thoroughly and have added my additional questions and concerns.

Reproducibility:

To reproduce results from the current manuscript, I used R version 4.3.3, which slightly differs from the version used by the authors (4.3.0). I reviewed results using both the datasets: "T1_ET.csv ,T1_TS.csv, T2_ET_Percept.csv, T2_TS_Percept.csv, IDS.csv" and code "Longitudinal_Uncertainty_OSF.R" uploaded in 2024 as well as the original files uploaded in 2023, including the datasets: "Time1_Imp.csv, T2_TS_memory.csv, T2_language_TOM.csv, T2_ET_memory.csv, T1_TS_memory.csv, T1_ET_memory.csv, Participants.csv" and code "Longitudinal_memory.R".

Major:

1. As a key point, the original data upload and code reproduce the path model results. However, there were slight differences in the subjects reported in the manuscript and those reproduced with the code and data provided on OSF with the 2024 timestamp. For example, there are 169 subjects after imputation for time 1 using the data frame "T1" from line 551 and 158 subjects after imputation for the Time 2 data, using the data frame "path" from line 549. The path model showed 97 females in the reproduced analyses, and 83 females at Time 2. The total length (number of subjects) of the integrated path

dataset was 183 subjects for the 2024 upload with supplementary analyses, versus the 176 reported in the manuscript and included in the first upload. This also resulted in some deviations for the average ages at each time point and the average difference between time points. Consequently, it would be helpful for the authors to notate any additional steps taken for others to be able to reproduce the results in the main body of the manuscript or that differ in the dataset for the additional supplementary analyses provided to address reviewer comments.

2. Although not preregistered, as noted by the authors in the rebuttal letter. It would be worthwhile to discuss the supplementary results provided in response to feedback from Reviewer 2, examining response latencies and gaze transitions. Even if the bulk of the analysis and results are included in the supplementary material, this data provides additional insight into not only the sample, but the authors' conclusions that "...eye-movements reflect information seeking, which may enable children to become attuned to their state of knowledge over time."

3. In their description of the present study (marked manuscript ls. 166-173), are the authors referring to the active and passive conditions from Leckey et al., (2024) described in the introduction? If so, it would be helpful and transparent to draw this connection for the readers unless these are different tasks from the participants from that study. This was unclear and would improve the ability to draw parallels between the work and understand how this study builds upon it (if this is the case). I understand that they are a subset of the participants, but the wording of this section had me questioning if the tasks differed.

4. I may have missed this information in the manuscript/rebuttal, but do the authors have any speculation about the lack of correlation between gaze transitions and memory accuracy at Time 1? It would be interesting to have this perspective in the manuscript to consider how the culmination of these behaviors working in tandem emerges.

Minor:

5. Additional models conducted and reported in the paper used to strengthen the argument of the authors' should have the full breadth of results reported in the supplementary analysis, with the caveat of overall poor fit still reported. This relates to the response to Reviewer 3, point 7.

6. As an extremely minor point, it would be good for the authors to specify that their standardized estimates come from lavaan using all path information to determine the standard estimates for the paths (i.e., std.all) versus a latent variable estimate (i.e., std.lv) that is additionally provided by this software.

7. Little context is provided to readers for how references to gaze transitions in the abstract should relate to metacognition and results (described in ls. 181-183 in marked up manuscript).

8. The phrasing "continuity in memory functioning" (l. 194) seems awkward to me, does this imply that performance will be equivalent across time-points?

9. Given this is not a change score model, can the authors provide an interpretation for the age difference correlation they report.

10. The wording of the part of the conclusion made in ls. 379 (starting in l. 377) in the marked-up document is unclear, "...toddlers' emerging ability to calibrate their assessment of memory states to confidence reports."

11. I would suggest one more tweak to Figure 3 to improve readability; namely, making sure parameters are not impeded by the lines.

12. For completeness, can the authors provide the model information in the Supplementary Material for the "... model in which ToM scores at Time 2 were replaced with scores that were conditionalized by accuracy in the control check question (Sobel & Austerweil, 2016). Overall fit and results for this model were virtually identical to the one reported above (robust $X^2(27) = 46.75$, $p = .011$, robust RMSEA = .06, and robust CFI = .89). (l. 316 – 320 in marked-up manuscript).

13. I suggest explicitly stating that the evaluation at age 2 predicts the abilities at age 3 for specificity (l. 463).

Version 2:

Reviewer comments:

Reviewer #4

(Remarks to the Author)

Manuscript ID: NCOMMS-23-61769B

Manuscript Title: Two-year-olds' Behavioral Indices of Memory Evaluation Predict the Emergence of Metamemory Monitoring One Year Later

I thank the authors for addressing each of my points and for the thoughtful explanation and approach to their pre-registration files. I defer to their decision about not adding speculation in the interpretation about the age difference result.

May 24, 2024

Reviewers' Comments

Reviewer #1 (Remarks to the Author):

This is a very nice study testing the relationship between behavioral manifestations of uncertainty monitoring during early childhood and later emerging abilities to provide metacognitive reports about memory accuracy. The findings are really interesting in that a longitudinal relationship is found between early information seeking / uncertainty monitoring and the later emerging ability to report about the accuracy of one's own memories on a (picture-based) confidence scale. It is also remarkable that gaze transitions were associated with confidence judgements at Time 2, a finding that parallels findings with adults but is still important because it confirms previous studies suggesting that gaze transitions might reflect confidence in early childhood. Overall, I applaud the authors for this really nice contribution to the literature.

We thank the Reviewer for this positive evaluation and for the helpful suggestions which we followed in order to further strengthen our manuscript.

Major:

1. To me the beginning of the abstract does not reflect the state of the art nor the aim of the study appropriately... It is also not very clear (e.g., how is uncertainty monitoring not a form of introspection)? I would suggest rewriting the first two sentences at least; what we don't know is how early behavioral manifestations of uncertainty monitoring (hereafter UM) relate to the development of more complex and abstract forms of metamemory reflected in reports (hereafter MR), and this is what the study tries (and manages) to test; at the moment it reads as if little was known about memory monitoring during early childhood at all, which is not correct.

We thank the Reviewer for this comment and apologize for the lack of clarity in our part. We have adjusted the first couple of sentences in our abstract for clarity (p. 2). Additionally, we would like to mention that although we do agree that there is previous work on early metacognition in other cognitive abilities (i.e., Dautriche et al., 2022; Lyons & Ghetti, 2011; Lyons & Ghetti, 2013; Marazita & Merriman, 2004), we do not believe that there has been a large amount of work on metamemory monitoring. To our knowledge, there have only been a few studies that attempted to examine the foundations of metamemory monitoring in children under 3 years of age (Gardier & Geurten, 2024; Goupil et al., 2016). Importantly, one novel aspect of our study is that is absent from previous studies is that we examined not only overt responses (i.e., memory decisions), but also behaviors occurring between a decision probe and a response (eye movement and response latency data), which provide new insight on how memory decisions are made in 2-year-olds. Moreover, the collection of the same data in 3-year-olds in

addition to confidence data allow us to link those implicit behaviors with overt uncertainty monitoring (i.e., degree of confidence difference as a function of response accuracy), thereby allowing for the assessment of implicit behaviors and confidence responses prospectively and concurrently.

2. It would be nice to either define precisely and/or avoid using the term “introspection” (cf also my remark above): this is a rather ambiguous term.

We agree that introspection is a broad term, but it is also one that most readers who are not familiar with metacognitive processes may understand to mean the process of evaluating a psychological process or state. Nevertheless, we do agree with the Reviewer that language precision is paramount. Thus, we edited the manuscript carefully to refer to metacognitive processes to increase precision of our language and retained the expression “introspection” only when we are seeking to situate this work more generally so that it can be readily consumed by non-experts (i.e., in the abstract/section headings).

3. The introduction lacks a description of computational models linking confidence and RTs in adults (e.g., see Pleskac & Busemeyer for an example). Many studies have been dedicated to this issue in adults, and could inform the fuzzy findings obtained in children (e.g., it is possible that the speed-accuracy tradeoff that governs the relationship between confidence and RTs is noisier for decisions that are more difficult for children); a similar point could be made for the discussion.

We thank the reviewer for this helpful suggestion. We have now included a brief description of these models in the Introduction (p. 6) and some additional detail in the Discussion (p. 18).

4. It would be good to specify briefly what the main measures are more at the beginning of the results section (e.g., what is metacognitive monitoring exactly). Similarly, the control task using novelty preferences is not presented and appears suddenly in the results section, this could be highlighted more at the stage of the hypotheses because it is a nice control?

We thank the Reviewer for this comment. We do agree that because the Results section occurs before the Methods section it can be confusing for the reader to determine what the main measures are. We have now added information where needed to clarify any measures for the reader (p. 10). We have also added in some information about our control task when we are discussing our hypotheses, making sure to mark it as an exploratory analysis because this specific analysis was not pre-registered (p. 8).

5. The contrast between the lack of findings relating ToM and language measures to MR as compared to the UM measures is striking! This has important theoretical implications and supports dual models of metacognition over mindreading models of metacognition, this could be emphasized more in the discussion (the theories are briefly mentioned in the

introduction but the findings are not contextualized within this theoretical debate in the discussion).

We thank the Reviewer for underscoring the importance of these findings and their suggestion. We have now added in more information about the debate surrounding the emergence of ToM and uncertainty monitoring in the Discussion (p. 19).

Minor:

6. 1.95: if I understand correctly this study concerns a subpart of the current dataset? If so it would be good to specify this here.

We thank the Reviewer for this comment and the opportunity to clarify. The data from Time 1 (memory performance in 2-year-olds) were reported in Leckey et al., 2024 as Experiment 1 and its direct replication reported in the Supplemental Materials of that manuscript. Here, we used all of the data from all children at Time 1 who were part of the longitudinal sample. A subset of children who were tested only on the replication of Experiment 1 (N=44) were not part of the longitudinal dataset and for this reason, they were not assessed at Time 2. We have added a sentence to make this clearer for the reader (p. 6).

Critically, our previously published work (Leckey et al., 2020; Leckey et al., 2024) on these data set has not investigated longitudinal memory findings or metamemory monitoring. This point is also discussed in response to Reviewer 2's Comment 1.

7. 1.239: I don't understand "despite being more accurate" here? I thought this result means that there is an association between gaze shifts and overall subjective confidence regardless of type1 and type2 accuracy (since this is entered as a covariate in the model)? This part of the results section is unclear...

We thank the Reviewer for this comment and apologize for our lack of clarity. The Reviewer is correct that there is a negative association between Time 2 gaze transitions and Time 2 average confidence (i.e., more gaze transitions are associated with lower decision confidence). What we were attempting to point out is that even though more Time 2 gaze transitions were associated with lower overall concurrent Time 2 confidence, they were also positively associated with Time 2 average accuracy, such that more gaze transitions were associated with higher Time 2 accuracy. Therefore, even though higher visual exploration of the test probes seemed to support accurate response selection, children may have found this increased visual inspection as a cue of lower confidence (as is found in older children and adults). For example, Selmecky and colleagues (2021) showed that 6- to 10-year-old children as well as adults showed higher accuracy rates, but reported lower confidence, for retrieval trials that required them to compare two test images more closely relative to retrieval trials in which such a close comparison was not encouraged.

We have now adjusted this sentence to reflect this explanation (p. 12).

8. 1.292-294 : “may indicate information seeking and evaluation, might allow more opportunities for toddlers to recognize differences in their memory states” and “From this perspective, these visual explorations need not to be themselves metacognitive” : these two sentences are contradictory ! I think you mean that it is not metamemory yet? Or perhaps you mean these are not metacognitive judgements/reports? But if there are evaluating and seeking information, they are doing something metacognitive... this wording is not really consistent with the reported findings that there is some specific longitudinal continuity.

We thank the Reviewer for this comment and welcome the opportunity to clarify. Engaging in more gaze transitions during decision-making is an indication that the toddler is seeking more information through visual exploration before making a decision. However, the toddler may not have conscious access to their need for additional information. From this perspective, the processes may not be overtly metacognitive (we define metamemory monitoring as conscious access to memory through explicit confidence ratings). However, continued experience with information seeking, including visual exploration, may provide support for the learning of the association between such behavior and the mental state that underlies it (e.g., lack of information). With this support, they may become able to use visual exploration of response options as a cue to uncertainty, eventually being able to report that uncertainty verbally. We have adjusted these sentences to convey this better to the reader (p. 14). We hope that this clarifies our point, but we welcome suggestions to clarify this further, if the reviewer deems it necessary.

9. Some recent articles are extremely relevant but not covered in the introduction (circa 1.71-80) – as argued by the authors, the longitudinal relationship between early UM and MR remained to be measured, but there were other indications already in the literature that the early manifestations of UM are metacognitive in nature rather than reflecting simpler mechanisms of risk avoidance (see for instance: Bazhydai et al., 2020; Dautriche et al., 2022; Lucca et al., 2020; Kuzyk et al., 2019).

We thank the Reviewer for pointing out these relevant articles. We have now added them into our introduction (p. 4).

10. Fig.3 caption lacks details (e.g., specifying what how betas and SE are depicted in the figure). It might be a good idea to use thicker lines for significant paths and lighter ones for non-sig paths, the figure is a little bit difficult to read at the moment...

We thank the Reviewer for this comment. We have now added more information into our figure caption about the betas and standard errors. We have also made the lines more distinguishable by thickening the significant paths and making the non-significant paths lighter (p. 42). We hope that this aids in the readability of the figure.

Reviewer #2 (Remarks to the Author):

This paper reports results from a longitudinal study examining the emergence of metamemory in toddlers. At time point 1, a large sample of two-year-olds completed a memory experiment. At time point 2, the now three-year-olds returned to complete an additional memory experiment and test of metamemory monitoring. The authors found that behaviors at time 1 predicted behaviors at time 2. Specifically, participants who had higher memory accuracy, faster response latencies, and more gaze transitions between response options at time 1 demonstrated better metamemory at time 2. The authors suggest that these results indicate that these ‘information-seeking’ behaviors at time 1 (i.e., gaze transitions) may be precursors to the emergence of metamemory.

This study has several notable strengths. I thought the question about the precursors to the developmental emergence of metamemory ability was very interesting and well-motivated, and the authors have collected an impressive dataset including a large number of participants at two timepoints. However, I have serious concerns about the robustness and replicability of the results, which I outline in more detail below. I also have concerns about the interpretation of the findings.

We appreciate the Reviewer’s assessment of our manuscript. We hope that we are able to alleviate any concerns that the Reviewer may have had with the changes to our manuscript and our responses to their comments below.

1. (a.) The authors state the study was preregistered, but the preregistration was created after (or on the same day) that the study was submitted, and (as noted in the preregistration) the data were already analyzed for conference abstracts prior to the preregistration. Thus, I don’t believe the authors should describe these analyses as “preregistered” as the “hypotheses” stated in the preregistration were determined after the data had already been analyzed. At the very least, the authors should note in the manuscript that the preregistration took place after data analysis. (b.) The primary reason I am concerned about the preregistration occurring after analyses is that the authors have a rich dataset with many measures, and it seems as though they have not appropriately corrected for multiple comparisons. Most reported p values are between .01 and .05, which suggests that the effects they do report may be small and unreliable. The authors have previously used other measures of memory and metamemory (e.g., novelty preferences, timecourse of novelty preference, parameters estimated from drift diffusion models) and so it is not clear why they chose the specific measures they did (i.e., gaze transitions and response latencies) or whether they tested other measures as well. I realize that asking the authors to replicate their findings in an additional sample is unreasonable given the fact that this data is longitudinal. But, particularly given the lack of evidence for significant metamemory monitoring in three-year-olds as a group, I’m worried that some of the reported relations may be noise.

We are grateful for the opportunity to clarify and will be responding to (a) and (b) consistent with the Reviewer's comments.

(a) We would like to first address the Reviewer's concern about our pre-registration. Our pre-registration was submitted on January 9th, 2023, which is almost a year prior to the submission of the manuscript to Nature Communications on December 12th, 2023. On the day we submitted the manuscript, we did not change anything in our pre-registration, but we uploaded our most recent data analysis script and datasheets to Open Science Framework. Therefore, that upload was not our pre-registration. We believe that the date of these uploads led the Reviewer to believe that we created the pre-registration then. Here is the link to the *preregistration* itself (<https://osf.io/9wz2m>, time stamped as 1/9/2023) and here is the link to the associated *project* where the final data and scripts are stored (osf.io/mz3xj, time stamped 12/12/2023). The OSF display is not always as clear as one would hope, which likely lead to this mistake. Like the reviewer, we would have also found it very inappropriate to submit a manuscript at the same time as the "preregistration" of its analysis. However, this is not what we did; we adhered to expected levels of rigor in our practices.

The conference abstracts we have submitted and presented (conference presentations listed at the bottom of this response) used parts of this larger dataset, but not the longitudinal analyses reported here. We never conducted any longitudinal analysis on memory or uncertainty monitoring about memory prior to completing the pre-registration. Similarly, we never presented the memory uncertainty monitoring data from Time 2 from this dataset at any conference, either before or after the pre-registration. We are happy to provide you with the content of the presentations listed below if you would like to double-check.

Here, however, we do recognize an error on our part. The section of the pre-registration mentioning conference presentations and preliminary longitudinal analysis does not specify that those analyses of uncertainty monitoring were completed on a different task dealing with perceptual uncertainty monitoring, which is the focus of a different pre-registration (<https://osf.io/ahde8>). We modeled the pre-registration for the analysis for the current paper (<https://osf.io/9wz2m>) after our previous pre-registration on analyses for the perceptual task (<https://osf.io/ahde8>), but unfortunately we failed to revise enough of the text in this specific section to specify that the longitudinal analysis we conducted concerned perceptual uncertainty monitoring, not the memory uncertainty monitoring task. We have now clarified this in the OSF page by linking all related projects together and explicitly posting what conference presentations and abstracts belong to each project.

Conference Presentations

Analysis on behavioral indicators associated with memory at Time 1

Leckey, S., Johnson, E., Ghetti, S. (2017). How do toddlers make memory decisions in the face of novelty preferences? 2017 Cognitive Development Society's Biennial Meeting, Portland, Oregon.

Leckey, S., & Ghetti, S. (2021). The benefit of visual exploration on toddlers' memory accuracy. 2021 Society for Research in Child Development Biennial Meeting, Virtual Conference.

Longitudinal analysis linking behavioral indicators associated with perception to uncertainty memory about perceptual decision at Time 2

Gonzales, C.R., Leckey, S., Selmeczy, D., Kazemi, A., Johnson, E., & Ghetti, S., (2021). Toddlers' behavioral indicators of uncertainty predict later metacognitive development. 2021 Jean Piaget Society Annual Meeting,

Leckey, S., & Ghetti, S. (2023). Longitudinal relations between eye-movement indices of evidence evaluation and the development of uncertainty monitoring from 3 to 5 years of age. 2023 Society for Research in Child Development Biennial Meeting

(b). With regards to the Reviewer's concern about our p-values, we would like to first clarify that because p-values are not an indication of effect size and should not be interpreted as such (see Dunkler et al., 2019), we do not comment on the magnitude of these values. We do report standardized beta-weights which are a better approximation for effect sizes (Hair, et al., 2017). Using our beta-weights, our effects would be considered to be small ($>.02$) to medium ($>.15$) effects.

Finally, the Reviewer also expressed concern about our choice of behavioral measures to examine in the manuscript. Our previous manuscript that was mentioned by the Reviewer did indeed use a variety of measures to describe how toddlers make memory decisions. However, not all of those measures had a direct theoretical basis to be related to the development of metamemory monitoring. We chose to only examine the behaviors of gaze transitions and response latencies because these two measures have been shown to be linked to uncertainty monitoring in older children and adults, as was outlined in our Introduction, in addition to them showing importance in the decision-making of toddlers in our previous study.

2. Related to my last point, the average metamemory score at time 2 is close to 0. Thus, it's not clear how many three-year-olds actually demonstrate above-chance metamemory. This is particularly concerning given the lack of the relation between gaze transitions at age 3 and metamemory at age 3, as well as the lack of the relation between response latencies at age 3 and metamemory at age 3. I thought the authors' speculative interpretation of the absence of the gaze transition effect was interesting and thoughtful, but not convincing without more data supporting it. It would be helpful for the authors to show a distribution of these metamemory scores, as well as to examine other potential behavioral differences between accurate and inaccurate trials to see if they differ more strongly in participants

with stronger metamemory. For example, my understanding is that the response latencies measure is a mean across all valid trials. Could the authors compute participants' response latencies separately for inaccurate vs. accurate trials? Perhaps participants who show a greater influence of accuracy on latency may also show stronger metamemory. Similarly, perhaps gaze transitions also differ for participants with better metamemory across accurate vs. inaccurate trials. If these relationships exist, it would help support the idea that some participants are in fact showing metamemory at age 3.

We thank the Reviewer for these comments and suggestions. There were 51 participants who demonstrated metamemory monitoring scores above 0 (i.e., they successfully monitored their memories based on confidence ratings). 3 years is expected to be an age when this ability is just beginning to be detectable and thus is not seen overall at the group level, which is why strong individual variability exists. This is consistent with the literature (Hembacher & Ghetti, 2014) in which researchers, using this same task, found evidence of metamemory monitoring at the group level in 4- and 5- year-olds but not in 3-year-olds. Here, we are capitalizing on this individual variability to identify factors promoting such an emergence. As suggested, we have included a scatterplot in our supplemental materials to show the distribution of metamemory scores across our sample. It has been included below for your reference.

At the Reviewer's suggestion, we examined more closely the relation between response latency, confidence, and accuracy at 3 years of age (i.e., Time 2). In the entire sample, although average response latencies did not appear to be concurrently associated with transitions and confidence at Time 2 (when evaluated in the overall longitudinal model; Figure 3), they were associated in the expected ways when examined in isolation. For example, average response latencies at T2 were significantly and positively associated with average gaze transitions at T2 ($r=.23$, $p=.018$), such that overall, those 3-year-olds who took longer to respond also engaged in more gaze transitions. Moreover, among 3-year-olds who showed evidence of metamemory monitoring (confidence for accurate trials *minus* confidence for inaccurate trials > 0), there was a correlation between metamemory monitoring and the degree to which children took longer to respond to inaccurate trials relative to accurate trials ($r=.27$, $p=.05$, $N=51$). In a multiple regression predicting metamemory monitoring from the difference in response latencies between inaccurate and accurate trials, memory accuracy, and age, this response latency difference score and accuracy independently predicted metamemory monitoring ($B=.32$, $p=.018$ and $B=.48$, $p=.008$, respectively). Thus, although gaze transitions at 3-years of age are generally associated with overall lower confidence, among children with stronger metamemory monitoring, the measure of response latency discriminability between accurate and inaccurate was also relevant. We have not included these additional analyses in the manuscript thinking that the reviewer wanted to see them for their own understanding of the results, but we would be happy to include them, if they think they are helpful.

As discussed in the manuscript, response latencies in 3-year-olds, however, are not associated with confidence in the entire sample, consistent with literature showing that for memory decisions, the use of response latency as a cue to metamemory monitoring continues to develop into late childhood, well after metamemory abilities are reliably observed (Ackerman & Koriat,

2011; Koriat & Ackerman, 2010). Further consistent with the idea that response latencies become more informative later in development, a portion of the children participating in this research were followed up and they showed higher confidence for accurate compared to inaccurate trials ($t(71) = 4.70, p < .001$) and slower response latencies for inaccurate trials compared to accurate trials ($t(71) = 3.80, p < .001$) when tested with this same task between the ages of 4 and 7 years.

Gaze transitions seem more informative about subjective confidence across the entire sample. The main longitudinal model reveals a concurrent association between gaze transitions and confidence, such that 3-year-olds who responded less confidently overall were more likely to show more gaze transitions ($b = -.19, p < .05$; Figure 3). We note that the association is primarily driven by inaccurate trials ($r = -.20, p < .05$ for inaccurate trials; $r = -.15, p = .11$ for accurate trials; Average gaze transitions at Time 1 was also associated with gaze transitions for inaccurate trials at t2, $r = .22, p < .04$).

We confirmed this pattern with a multi-level approach to examine data at the trial level. We conducted a multilevel model predicting response latencies and gaze transitions from trial level confidence ratings. We found that for trials in which the children selected “not so sure” as their confidence rating, gaze transitions were significantly higher compared to trials in which children selected “really sure” as their confidence rating, $b = .16, z = 2.56, p = .011$. This indicates that gaze transitions informed decision confidence judgments at Time 2, even though average gaze transitions at Time 2 were not related to children’s ability to differentiate metamemory monitoring ability (i.e., the ability to differentiate between accurate and inaccurate responses with their confidence).

We do not propose that children are aware of their eye-movements and that they use this insight to guide their confidence reports. Instead, we propose that eye-movements reflect information seeking, which may enable children to become attuned to their state of knowledge over time (as reflected by longitudinal relation between gaze transitions at time 1 and uncertainly monitoring at Time 2) and may capture additional cues that children may begin to gain an awareness of (e.g., effort to find an answer) that does not necessarily map directly on time to respond (as reflected by concurrent association with average confidence at Time 2).

Overall, consistent with the reviewer’s suggestion, we do observe that Time 2 response latencies and gaze transitions are informative for confidence rating in the expected directions for those children who engage in effective metamemory monitoring at Time 2. We opted not to report all of these additional results in the manuscript given that these were not pre-registered and are not representative of the sample as whole. However, if the reviewer would like us to include it in our supplementary materials as follow-up exploratory analyses, we would be happy to do so.

3. I think the idea that visual exploration at time 1 predicts metamemory at time 2 is very interesting, but I found the gaze transition variable hard to interpret, particularly because the relation did not hold when looking only at the time 2 data. In prior papers, the authors have reported that gaze transitions or more distributed patterns of attention are related to memory accuracy. They found that all toddlers demonstrated an initial novelty preference, but toddlers who were able to inhibit this novelty preference and focus on the goal of identifying the old image (i.e., make a gaze transition to the old image) made more accurate memory responses. So it seems like all three time point 1 predictors (higher memory accuracy, faster response latencies, more gaze transitions) are associated with better memory performance, and that better memory performance at time 1 may predict metamemory at time 2. It is not clear to me that any of these measures are necessarily direct precursors to metamemory ability. It would be helpful for the authors to report whether gaze transitions and latencies at time 1 predicted time 2 memory accuracy. It was not clear to me if these relations were tested, but if they do, then this may suggest that they are stronger indices of memory ability (vs. metamemory) and that memory ability itself predicts the emergence of metamemory.

We thank the Reviewer for this comment. We agree with the Reviewer's reading of our previous paper in which we showed that 2-year-olds who were able to divert their gaze away from the novel stimulus and examine the images more equally, were able to make more accurate memory responses, indicating that the ability to transition away from the novel item increases memory performance.

In the main longitudinal model (Figure 3), memory accuracy at both Time 1 and Time 2 were included in order to account respectively for the longitudinal and concurrent relation to metamemory monitoring. As the Reviewer suggested, Figure 3 indeed shows that Time 1 memory accuracy predicts metamemory monitoring at Time 2. As such, average transitions at Time 1 predict metamemory monitoring at Time 2 above and beyond the effect of memory accuracy at Time 1 and Time 2. The path from memory accuracy at Time 2 to metamemory monitoring at time 2 was not significant. In addition, the model accounts for all of the concurrent associations between response latencies, gaze transitions, and memory accuracy. However, the Reviewer is correct that we did not test for the additional paths from response times at Time 1 and gaze transitions at Time 1 to memory accuracy at Time 2. We initially did not do so because memory accuracy was already included and considered the additional longitudinal paths across all the predictors unnecessary for the current scopes. However, when we did so, we found that those paths did not significantly predict memory accuracy at Time 2 and the paths from gaze transitions and response latencies at Time 1 to Time 2 remain unchanged (Time 1 variables did not predict Time 2 variables). If the reviewer felt it would be helpful to report this additional model, we could include this in the supplementary materials.

More minor points:

1. I'm a little confused about the prior publication of the Timepoint 1 data. The paper that has been published with the Timepoint 1 data (I think) has 206 participants (perhaps I am looking at the wrong Leckey et al., 2023?). Does this study include only a subset of them?

We apologize for the confusion. As we discussed in a point raised by the other Reviewer (Reviewer 1, Question 6), the data from Time 1 (memory performance in 2-year-olds) were reported in Leckey et al., 2023 as Experiment 1 and its replication. Here, we used the data from Time 1 from all children across Experiment 1 and its replication that were also part of our longitudinal sample. However, a subset of children (N=44) were recruited to take part only in the replication of Experiment 1 to achieve the desired sample size for the replication of the study, which meant that they only completed the eye tracker version of the task and not the touchscreen version. Therefore, the Reviewer is correct that this current study does not include all children in the Leckey et al., 2023 study. We did conduct a model with these participants included and all findings remained the same. Therefore, we elected to exclude these participants.

2. It is somewhat difficult to interpret Figure 3 (because it's not always clear what labels correspond to what lines) but it looks as though there are strong effects of age that the authors should report in the main text.

We thank the Reviewer for this suggestion. We have edited Figure 3 in order to make the significant paths and their information clearer for the reader (p. 42). We have additionally added in some discussion about the relation between age and our variables in the manuscript (p. 12).

Reviewer #3 (Remarks to the Author):

The current longitudinal study examined the development of toddler and preschool children’s early memory and indicators of metacognition to predict later memory and decision confidence. Findings showed that early gaze transitions between responses, shorter response latencies, and memory accuracy at Time 1 predicted metamemory monitoring at Time 2. Alternate possibilities such as novelty preference and the role of mental state language and theory of mind were examined but ruled out as contributing to the current findings. Findings illuminate some of the early building blocks of emerging memory introspection.

This study presents important findings for the memory and metacognition fields. The longitudinal evidence is powerful and allows for testing the emergence of these abilities and examine their predictive power for later emerging abilities. The study has been rigorously conducted and methods are described in sufficient detail (although see some comments below), the analyses are appropriate, and the results are interpreted accurately. I especially appreciate that the authors took the care to rule out some alternative possibilities surrounding the role of novelty preference as well as mental state language and theory of mind. Given the strength of this study and manuscript, I just have relatively minor comments that are aimed at improving clarity and readability of the manuscript.

We thank the Reviewer for the favorable review of our manuscript.

Comments:

1. On page 3, line 64: I wasn’t clear what “asking for a cue” meant after an inaccurate response. Can this be clarified or expanded to make the meaning clearer?

We thank the Reviewer for this comment. In that specific study (Geurten & Bastin, 2019), after making their original selection of which image they had seen before, the researchers gave the children an opportunity to choose whether they wanted to receive a cue about the correct answer or not. When their original answer was incorrect, children said that they wanted a cue more often than when their original answer was correct. We have now adjusted our explanation of this study in the manuscript to make this clearer for the reader (p. 3).

2. In the description of Leckey and colleagues (2023) on page 5, it was not at first clear to me that looking preferences were also measured in the active condition, until I read the sentence about the high performing group on line 107. Maybe it’s worth saying this explicitly?

We thank the Reviewer for this suggestion. In the explanation of this study, we now explicitly say that looking preferences were collected in both the Active and Passive conditions (p. 5).

3. I suggest that the author put the instructions to children “Now we are going to help Julia

find her drawings” and “Which picture did Julia show you before? Only one is Julia’s” closer together in the text, as I wondered at first whether children only received the first part of the instruction which seems quite vague without the follow-up question before each trial.

We thank the Reviewer for this suggestion. We have now adjusted the description of our task to have the question that the toddlers were asked earlier (p. 23).

4. I found the phrase “The vocabulary score was calculated by taking the average of the 100 words that the parents checked off” a bit confusing. Did both parents complete the MacArthur-Bates Inventory and so you averaged the two parents scores? Or was the score just the number of words that one parent had selected/checked off. The reference to “average” threw me off here.

We thank the Reviewer for this comment. To clarify, only one parent completed the MacArthur-Bates survey. The completion of this survey requires the parent go through a list of 100 words and mark the ones the child was able to say. A score of 1 was given to the words that were checked off and a score of 0 was given to the words that were not checked off. For each child, the scores were added up and then we divided that sum by the number of words that there were, 100. Therefore, it was technically an average score. However, we do agree that our wording may have been confusing. Therefore, we have slightly changed the sentence to use proportion instead of average as this wording may be clearer to readers (p. 26).

5. On page 26, line 581: did the authors mean that children were told that they did not see the seahorse before? If children were incorrect, it seems like they should have been told that they did not see the seahorse previously.

We thank the Reviewer for this comment. These were example statements of what was said during the practice trials if the children were correct and if they were incorrect, imagining that the seahorse was the correct image that was supposed to be chosen. If the child correctly chose the seahorse, they were told that they were correct and that we chose the seahorse because we remembered that we saw the seahorse before. However, if the child chose the other picture that the seahorse was paired with, they were told that they actually saw the seahorse before. Therefore, the incorrect example statement is correct, because they were supposed to have chosen the seahorse, but did not. We hope that this clears up any confusion.

6. It would be nice to include the 3-point pictorial confidence scale as a figure, or perhaps in supplemental materials.

We thank the Reviewer for this suggestion. The 3-point pictorial confidence scale is included in Figure 1b. We do agree that it may have been a little hard to see, therefore, we have adjusted the figure to allow the reader to see the scale easier (p. 42).

7. Were children asked control or comprehension check questions on the explicit false belief task (e.g., where are the mittens really?). If so, could accuracy on these questions be reported.

We thank the Reviewer for this question. Participants were indeed asked a comprehension check question in the False Belief Task. After laying out the set-up, the children were asked where the character would search for the item, then they were asked where the item actually was. We have now included this information in the manuscript and reported the accuracy on the comprehension check question (p. 31). In light of the reviewer's important observation and consideration of alternative coding schemes for false-belief tasks (Sobel & Austerweil, 2016), we recoded our data such that we gave credit for passing the false-belief task only to children who responded correctly to the control check question. The results for this model were virtually identical to the model reported in the text. Fit indices were similar, suggesting that this new model also had a poor fit (robust $X^2(27) = 46.75$, $p = .011$, robust RMSEA = .06, and robust CFI = .89). Similar to the model in the text, the paths between Time 1 "I don't know" ratings and Time 2 metamemory monitoring and between Time 2 ToM and Time 2 metamemory monitoring were not significant ($\beta = .07$, SE = .02, $p = .400$; $\beta = -.23$, SE = .17, $p = .068$). However, we did find a significant path from Time 2 mental state language to Time 2 metamemory monitoring that was not significant in the previous model, $\beta = .20$, SE = .05, $p = .048$. Considering that the model fit was overall poor, we elected not to report the full model in the text and only mention that we conducted this separate model that showed similar results (p. 14). However, if the Reviewer deems it important to include the full results of the model, we would be happy to include it in the Supplemental Materials.

8. Please ensure Results section is written in past tense (e.g., "metamemory monitoring at Time 2 is not..." – should be "was not"; "but there is substantial individual variation" should be "there was substantial...").

We thank the Reviewer for pointing out our errors in tense. We have now gone through the results section and ensured that it has been written in the past tense.

December 11, 2024

Reviewer Comments

Reviewer #1 (Remarks to the Author):

The authors have successfully addressed all of my comments.

We thank the Reviewer for all of the helpful comments from our previous manuscript version.

Reviewer #2 (Remarks to the Author):

In my original review, I noted the interesting question the study addresses but raised concerns about its replicability. The authors have now assuaged some of my concerns, though I believe some further edits to the text are needed to clarify which analyses were preregistered.

We thank the Reviewer for the additional suggestions and we hope that we can further alleviate any remaining concerns that the Reviewer may still have with this revision.

1. First, I apologize for mistakenly interpreting the date of the preregistration. I now see that the preregistration was submitted well in advance of the manuscript's submission. It does appear as though the primary path model of interest was preregistered (though again, the preregistration was submitted after preliminary analyses were conducted, so I remain skeptical of its utility. I will take the authors' word that they did not conduct analyses of the longitudinal data until after preregistration).

We thank the Reviewer for this comment and the chance to further clarify our pre-registration timing and wording. We would like to reiterate that at no point were preliminary analyses conducted on the longitudinal analyses reported for this manuscript, and thus feel that we must insist on the utility of our pre-registration. The "preliminary analyses" that are mentioned in the pre-registration refer to analyses for conference abstracts that we conducted on a different part of the larger data set. Those conference abstracts were listed in our previous response and did not include any of the dependent measures used in this longitudinal analysis. To the extent that memory measures were involved, previous analyses only focused on data collected at 2 years of age and not the longitudinal follow-up. This process is, of course, quite common, such as the practice of pre-registering longitudinal analyses after the publication of manuscripts reporting results from the first assessment point or longitudinal analyses concerning other variables (e.g., the many pre-registered plans from the ABCD study). For clarity, we linked the OSF pages for all analyses that have been conducted and published with other aspects of this larger data set. We hope that this has helped alleviate the Reviewer's concerns.

2. That said, some of the analyses reported in the manuscript seem to deviate from those that were preregistered, and the authors should note these in the text. The preregistration

specifies separate analyses for mental state language and theory of mind, but those these appear to be combined in the manuscript.

We thank the Reviewer for this comment. We do agree that in the pre-registration we listed the analyses for mental state language and theory of mind separately, as this went along with our hypotheses listed at the beginning of the pre-registration. When we conducted the analyses initially, we did indeed conduct separate path models for mental state language and theory of mind. However, the results for these models were the same as the model that includes both predictors and is reported currently in the manuscript. Therefore, in order to keep the manuscript as concise as possible we elected to use the combined model in the final manuscript version. However, we agree that reporting both models increase transparency and correspondence with the pre-registration. Therefore, we have now included these separate models in the Supplementary Materials (Supplementary Results 3) and have no included information about this particular deviation from the pre-registration (p. 14). This way, we maintain concision in the main manuscript, but we also explicitly note the deviation from the pre-registration.

3. In addition, the manuscript models include more variables than those specified in the preregistration. This is relatively minor as these results were not the primary ones of interest in the manuscript, but the authors should note this deviation.

We thank the Reviewer for the opportunity to clarify. There was one missing variable from our pre-registration and this was overall vocabulary at Time 1. All other control variables are listed, both in the “Variables” section under the “measured variables” and “indices” headers and in the “Analysis Plan” section under the “statistical models” header. We did neglect to list overall vocabulary in these sections. However, we deemed it important to include in order to control for the possibility that any findings surrounding mental state language were only due to the children’s vocabulary size. We have now noted this one deviation in our manuscript (p. 29).

4. It is also not clear why the researchers chose to impute missing accuracy scores from participants who did not complete both versions of the task (an analytic decision not specified in the preregistration).

We thank the Reviewer for this comment. We agree with the Reviewer that we neglected to describe our imputation procedure for accuracy scores in our pre-registration. There is consensus in the field that dealing with missing data with imputation techniques is less biased and provides a more accurate representation of the data than using a listwise or pairwise deletion procedures that would result from conducting the longitudinal analysis without using imputation techniques (Zhang, 2016). In retrospect, we should have stated the selection of this approach in the pre-registration. We have now added this deviation and reasoning into the manuscript (p. 27).

5. I am still left wishing that the authors could replicate their complex (and small) effects in an additional sample. I recognize this is not a reasonable request given the scope of data collection, and so I don’t believe this concern should prevent publication of the manuscript, particularly given the positive comments from the other reviewers. But I would encourage the authors to split their data into discovery and replication samples in the future.

We thank the Reviewer for this comment. We do agree that having a “discovery” and “replication” sample is ideal. Indeed, we have used this approach when the hypothesized effect size of an experimental manipulation was large enough such that our resources allowed for split data collection in a first experiment for discovery and a second experiment as a direct replication. We were able to successfully do this with our Time 1 data (Leckey et al., 2024). However, realistic longitudinal effects may be smaller and, unfortunately, our funding did not allow for including a discovery and replication sample for the purpose of longitudinal analyses. In the future we will continue to try our very best to seek out resources that would make it possible to take this approach for smaller effect sizes within the context of longitudinal analyses.

Reviewer #3 (Remarks to the Author):

I am satisfied with the revisions that the authors have made in response to my concerns and the other reviewers' concerns. I recommend accepting this manuscript.

We thank the Reviewer for all of the helpful comments from our previous manuscript version and for the positive evaluation of our manuscript.

Reviewer #4 (Remarks to the Author):

Manuscript ID: NCOMMS-23-61769A-Z

Manuscript Title: Two-year-olds' Behavioral Indices of Memory Evaluation Predict the Emergence of Metamemory Monitoring One Year Later

Summary: While I did not serve reviewer on the initial submission, the current manuscript is a revision addressing a set of concerns from the initial reviewers. I first reviewed the manuscript, then viewed the original set of concerns and rebuttal letter. The manuscript addresses a novel and interesting topic of identifying behavioral precedents at age 2 to emerging metamemory monitoring at age 3. The authors have thoughtfully addressed this question, using longitudinal data and path modeling to simultaneously assess questions of interest. Further, they consider plausible alternatives and give space for these alternative theories to play a role in other aspects of cognitive development. Overall, I am impressed by the authors' work. I think the authors have addressed the prior set of reviews quite thoroughly and have added my additional questions and concerns.

We thank the Reviewer for this positive evaluation and for the helpful suggestions which we followed in order to further strengthen our manuscript.

Reproducibility:

To reproduce results from the current manuscript, I used R version 4.3.3, which slightly differs from the version used by the authors (4.3.0). I reviewed results using both the datasets: “T1_ET.csv ,T1_TS.csv, T2_ET_Percept.csv, T2_TS_Percept.csv, IDS.csv” and

code “Longitudinal_Uncertainty_OSF.R” uploaded in 2024 as well as the original files uploaded in 2023, including the datasets: “Time1_Imp.csv, T2_TS_memory.csv, T2_language_TOM.csv, T2_ET_memory.csv, T1_TS_memory.csv, T1_ET_memory.csv, Participants.csv” and code “Longitudinal_memory.R”.

Major:

1. As a key point, the original data upload and code reproduce the path model results. However, there were slight differences in the subjects reported in the manuscript and those reproduced with the code and data provided on OSF with the 2024 timestamp. For example, there are 169 subjects after imputation for time 1 using the data frame “T1” from line 551 and 158 subjects after imputation for the Time 2 data, using the data frame “path” from line 549. The path model showed 97 females in the reproduced analyses, and 83 females at Time 2. The total length (number of subjects) of the integrated path dataset was 183 subjects for the 2024 upload with supplementary analyses, versus the 176 reported in the manuscript and included in the first upload. This also resulted in some deviations for the average ages at each time point and the average difference between time points. Consequently, it would be helpful for the authors to notate any additional steps taken for others to be able to reproduce the results in the main body of the manuscript or that differ in the dataset for the additional supplementary analyses provided to address reviewer comments.

We thank the Reviewer for taking the time to download our data and check our scripts and reporting. We would like to note that the two sets of data that the Reviewer downloaded were not both relevant for this current study and manuscript. The discrepancies noted by the Reviewer concerns data that are not relevant for the current article (e.g., variables that include “Percept” in the name). The dataset uploaded in 2023 is the current and correct data for this manuscript (all the CSVs and R scripts that have “memory” in the title). The data uploaded in 2024 concerns another manuscript (Leckey, Gonzales, Selmecky, & Ghetti, 2024). Therefore, the integrated path dataset utilizing the 2024 datasheets and scripts (“T1_ET.csv ,T1_TS.csv, T2_ET_Percept.csv, T2_TS_Percept.csv, IDS.csv” and code “Longitudinal_Uncertainty_OSF.R”) will include a total of 183 participants with 83 females participating at Time 2. These are the numbers reported by the Reviewer, but again these are not the variables examined here.

However, if the uploaded data from 2023 and associated script are used (“Time1_Imp.csv, T2_TS_memory.csv, T2_language_TOM.csv, T2_ET_memory.csv, T1_TS_memory.csv, T1_ET_memory.csv, Participants.csv” and code “Longitudinal_memory.R”), the number of participants will be 176, the number of participants included in the current study. This confusion is caused by the way that OSF handles the stored items when you link projects together. In response to the confusion surrounding our pre-registration from the last round of reviews, and to be transparent for all future readers, we decided to link together all of the projects that deal with the data from the larger study that this sample is taken from. When we did this, all of the data and variables from those studies were added to the “files” storage section, which is not ideal. Any of the main headers for the file storage section that has a linked chain next to it are from the linked studies and not the current study that the reviewer is viewing. The current studies data is

under the header that has the cube next to it. We would also like to mention that we will be uploading new datasheets and script with the new revision with changes based on all of the Reviewer comments and requests. Therefore, the correct datasheets will now have a December, 2024 date but will still all have “memory” in the title.

We hope that this explanation alleviates any confusion and concerns, about noted differences in our sample numbers and variable averages.

2. Although not preregistered, as noted by the authors in the rebuttal letter. It would be worthwhile to discuss the supplementary results provided in response to feedback from Reviewer 2, examining response latencies and gaze transitions. Even if the bulk of the analysis and results are included in the supplementary material, this data provides additional insight into not only the sample, but the authors’ conclusions that “...eye-movements reflect information seeking, which may enable children to become attuned to their state of knowledge over time.”

We thank the Reviewer for this suggestion. We have now included these analyses in the Supplementary Materials (Supplementary Results 1) and have additionally discussed them, albeit briefly, in the main manuscript, both in the Results section (p. 12) and the Discussion section (p. 17).

3. In their description of the present study (marked manuscript ls. 166-173), are the authors referring to the active and passive conditions from Leckey et al., (2024) described in the introduction? If so, it would be helpful and transparent to draw this connection for the readers unless these are different tasks from the participants from that study. This was unclear and would improve the ability to draw parallels between the work and understand how this study builds upon it (if this is the case). I understand that they are a subset of the participants, but the wording of this section had me questioning if the tasks differed.

We thank the Reviewer for this suggestion. The Reviewer is indeed correct that the tasks we are referring to are the Active and Passive conditions from the Leckey et al., (2024) study. We have now explicitly stated this in the manuscript (p. 8).

4. I may have missed this information in the manuscript/rebuttal, but do the authors have any speculation about the lack of correlation between gaze transitions and memory accuracy at Time 1? It would be interesting to have this perspective in the manuscript to consider how the culmination of these behaviors working in tandem emerges.

This is an excellent question. We are happy to offer a speculation. At Time 1, 2-year-olds show strong novelty preferences which are difficult for them to inhibit. Even though there are clear individual differences in the degree to which 2-year-olds visually explore and evaluate the two response options (and these individual differences in gaze transitions are associated with the response boundary parameter in a drift diffusion model, indicating that toddlers required more evidence before they made their decision), novelty preference drives response selection in the

majority of 2-year-olds. Indeed, we experimentally demonstrated that re-directing attention away from the novel item and toward the target was important to engage in accurate retrieval. Our speculation is that the fact that novelty preferences drive response selection in the majority of 2-year-olds (i.e., 2-year-olds tend to choose what they spend more time looking at) may obscure the possibility of finding a positive association between accuracy and gaze transitions at Time 1. We have added a brief discussion surrounding this point into the Discussion section (p. 18).

Minor:

5. Additional models conducted and reported in the paper used to strengthen the argument of the authors' should have the full breadth of results reported in the supplementary analysis, with the caveat of overall poor fit still reported. This relates to the response to Reviewer 3, point 7.

We thank the Reviewer for this suggestion. We have now included the full model and results in the Supplementary Materials (Supplementary Results 4).

6. As an extremely minor point, it would be good for the authors to specify that their standardized estimates come from lavaan using all path information to determine the standard estimates for the paths (i.e., std.all) versus a latent variable estimate (i.e., std.lv) that is additionally provided by this software.

We thank the Reviewer for this suggestion. We have now included this information in the discussion of our analysis plan (p. 35).

7. Little context is provided to readers for how references to gaze transitions in the abstract should relate to metacognition and results (described in ls. 181-183 in marked up manuscript).

We thank the Reviewer for pointing out this oversight. We have now adjusted our abstract to include some additional, brief, information surrounding gaze transitions and their connections to metacognition (p. 1).

8. The phrasing “continuity in memory functioning” (l. 194) seems awkward to me, does this imply that performance will be equivalent across time-points?

We thank the Reviewer for pointing out this issue with clarity. With this statement we were referring to the possibility that memory functioning may be related across time points (i.e., higher memory abilities at Time 1 predicting higher memory abilities at Time 2), not equivalent because we know that memory abilities develop throughout childhood. We have slightly adjusted this sentence in order to make this clearer for the reader (p. 9).

9. Given this is not a change score model, can the authors provide an interpretation for the age difference correlation they report.

We thank the Reviewer for this comment. There was some variability in time duration between the Time 1 Assessment and the Time 2 Assessment. This variation between time points is captured by the Age Difference variable. The correlations with this variable suggest that those children who were tested after a longer amount of time exhibited higher gaze transitions and memory accuracy. Given that age at the beginning is accounted for, we interpret this finding as suggesting a developmental trend (i.e., better ability associated with more time since first testing). We are reluctant to include an interpretation of this finding in the interest of space since it is not central to our hypotheses and we use age difference as a control variable (to account for slight variation in test timing), but we will include this if deemed necessary.

10. The wording of the part of the conclusion made in ls. 379 (starting in l. 377) in the marked-up document is unclear, "... toddlers' emerging ability to calibrate their assessment of memory states to confidence reports."

We thank the Reviewer for this comment. With this statement we were referring to the ability to match the accuracy of their memory decision with a confidence level (i.e., higher confidence ratings for accurate trials compared to inaccurate trials). We have now slightly changed the wording of this sentence to ensure clarity for the reader (p. 18).

11. I would suggest one more tweak to Figure 3 to improve readability; namely, making sure parameters are not impeded by the lines.

We thank the Reviewer for this comment. We have now adjusted the figure by having the parameters with a white background when they are over the path lines and bringing them all the way to the front when they are on top of path lines (p. 57). We hope that this helps with visibility in the figure.

12. For completeness, can the authors provide the model information in the Supplementary Material for the "... model in which ToM scores at Time 2 were replaced with scores that were conditionalized by accuracy in the control check question (Sobel & Austerweil, 2016). Overall fit and results for this model were virtually identical to the one reported above (robust $X^2(27) = 46.75$, $p = .011$, robust RMSEA = .06, and robust CFI = .89). (l. 316 – 320 in marked-up manuscript).

We thank the Reviewer for this suggestion. We have now added this model into the Supplementary Materials (Supplementary Results 4).

13. I suggest explicitly stating that the evaluation at age 2 predicts the abilities at age 3 for specificity (l. 463).

We thank the Reviewer for this suggestion. We have now changed the wording of this sentence to explicitly state "age 3" instead of "preschoolers" (p. 22).